# The actin module of endocytic internalization in *Aspergillus nidulans*: A critical role of the WISH/DIP/SPIN90 family protein Dip1

Marisa Delgado[ID]°, Mario Pinar[ID]°, Paula Polonio[ID]¤a, Sergio Fandiño[ID]¤b, Eduardo A. Espeso, Miguel A. Peñalva[ID]*

Department of Molecular and Cellular Biosciences, CSIC Centro de Investigaciones Biológicas Margarita Salas, Madrid, Spain

° These authors contributed equally to this work.
¤a Current address: Department of Biophysical Chemistry, CSIC Instituto de Química Física Blas Cabrera, Madrid, Spain
¤b Current address: Department of Animal Health, Faculty of Veterinary, Universidad Complutense, Madrid, Spain
* penalva@cib.csic.es

## Abstract

Using fluorescent protein-tagged F-actin reporters we studied the actin cytoskeleton in *Aspergillus nidulans*. F-actin probes labeled endocytic patches, contractile actin rings and the Spitzenkörper (SPK), but not exocytic cables generated by the SPK-associated formin, illuminated only by tropomyosin. The SPK actin mesh contains tropomyosin and capping protein, but not fimbrin or Arp2/3, showing that it does not involve branched actin. Arp2/3 and fimbrin are recruited to endocytic patches at the end of their lifecycle, staying in them for 12–14 sec, coinciding with the burst of branched actin polymerization that powers vesicle internalization, whereas verprolin stays only during the first half of this actin phase. Hyphal growth requires endocytic recycling, which we exploited to assess the efficiency of endocytosis following genetic interventions. Ablation of SlaB^Sla2, Arp2/3, cofilin and fimbrin is lethal, whereas that of Srv2, verprolin and capping protein are debilitating, with the lifetime of actin in mutant patches roughly correlating with the extent of growth and endocytic defects. An outstanding problem is the origin of seed filaments required to prime Arp2/3 during endocytosis. Actin patches associate with cortical cables that give rise to long distance-moving "actin worms" that are different from tropomyosin-containing cables emanating from the SPK. Cables and worms are dependent on formin, yet inactivation of formin does not affect the F-actin patch lifecycle, arguing against formin playing an endocytic role. Ablation of the WISH/DIP/SPIN90 protein Dip1 priming Arp2/3 for the synthesis of linear actin delocalizes the endocytic machinery and severely impairs, but does not preclude, endocytosis. This establishes the existence of Dip1-independent mechanism(s) that synthesize seed filaments. Our data negate the possibility that this alternative mechanism results from a priming role of formin

**Data availability statement:** All relevant data are within the manuscript and its Supporting information files. The values used to calculate statistical parameters are all displayed in the corresponding figures.

**Funding:** MAP and EAE were supported by Grants TED2021-129607B-I00 funded by the Ministerio de Ciencia, Innovación y Universidades/Agencia Estatal de Investigación and the "European Union NextGeneration EU/PRTR" program, and by Grant PID2021-124278OB-I00 funded by the Ministerio de Ciencia, Innovación y Universidades/ Agencia Estatal de Investigación and "ERDF A way of making Europe". MD was supported by a FPI Grant PRE2022-102144 funded by the Ministerio de Ciencia, Innovación y Universidades/Agencia Estatal de Investigación and by "ESF Investing in your future". The funders did not play any role in the study design, data collection and analysis, decision to publish, or preparation of the manuscript.

**Competing interests:** The authors declare no competing interests.

that is unmasked in *dip1Δ* cells, but do not exclude that cofilin-mediated filament severing could produce seed microfilaments for Arp2/3, as suggested for *Schizosaccharomyces pombe.*

## Author summary

Filamentous fungi have a deep impact in our lives as friends and foes. Certain species are used as cell factories for the production of proteins or biopharmaceuticals. In contrast, phytopathogenic species cause important losses in crops whereas those able to infect humans represent serious risks for global health. Filamentous fungi form tubular cells, denoted hyphae, that grow by apical extension. This requires the coupling between exocytosis and endocytosis in the so-called endocytic recycling pathway, which is needed, for example, to maintain the polarization of enzymes which synthesize the cell wall as growth proceeds. Remarkably, detailed studies on endocytosis in filamentous fungi are wanting. Here we report the characterization of the endocytic pathway in the genetic model *Aspergillus nidulans*, a filamentous ascomycete which is well-suited for genetic manipulation and *in vivo* fluorescence microscopy. Our study demonstrates that endocytosis is essential for filamentous fungal life and provides significant insight on how F-actin powers the internalization of endocytic vesicles, including an important physiological role of Dip1, a protein required to provide seed filaments for the formation of F-actin branching networks. Importantly, as yet unidentified Dip1-independent mechanisms that synthesize these seeds must exist, opening new avenues for future research.

## Introduction

Pre-synaptic buttons of neurons and hyphal tip cells of filamentous fungi share a distinct subcellular organization. Both are dependent on compensatory endocytosis to recycle membrane and associated proteins that fuse with the plasma membrane at the cellular pole targeted by exocytosis. In synapses, exocytosis occurs at a specialized zone, denoted the active zone, and endocytosis takes place in an area that surrounds that of the active zone [1–3]. In the filamentous ascomycete *Aspergillus nidulans* exocytosis is targeted to the apex, and compensatory endocytosis takes place in a region subtending the apical dome [4–6]. Endocytic recycling is crucial for the filamentous mode of life because cell-wall modifying enzymes that are integral membrane proteins polarize by this mechanism [7].

In spite of its basic and applied importance, endocytosis in filamentous fungi has been insufficiently studied. In addition to the intrinsic interest of understanding the fungal mode of life, studies of endocytosis as such are facilitated by three important facts [8]. One is that *Aspergillus nidulans* is a genetically amenable and ideally suited for live microscopy. Second, as endocytosis is essential, mutations impairing this process even to a relatively minor extent translate into a growth phenotype that can

be scored by simple growth tests. Third, as integral membrane proteins are efficiently captured by the subapical endocytic collar, mutations decreasing this efficiency permit the unrestrained diffusion of cargoes away from the hyphal tip, thereby providing a highly sensitive test to measure endocytosis [7,9,10].

Across the eukaryotic realm, endocytosis occurs in a specialized class of structures –actin patches–, in which endocytic pits develop into cargo-loaded vesicles that are internalized in a process that is powered by a burst of F-actin polymerization involving the Arp2/3 complex [11]. Since the pioneering work of Drubin's and other laboratories, a general model has emerged by which a score of endocytic proteins have been ordered in a pathway that is highly conserved [12–14]. Even though the details of this model have been worked out with notable detail, our understanding of certain aspects of this pathway is insufficient. For example, because the Arp2/3 complex catalyzes the formation of branching actin only on a preexistent actin filament, the proteins required to synthesize these seed filaments are still a subject of debate [15–18].

The process of endocytosis has been studied in filamentous fungal model organisms such as *Neurospora crassa* and *Aspergillus nidulans* [6,19–26]. However, most of these studies have been purely descriptive and attempts to use the experimental advantages of these fungi to gain insight into the process of endocytosis itself have been scarce. Here we exploit the experimental advantages of *A. nidulans* to study the roles of several endocytic proteins. Deletion alleles of key genes either resulted in lethality or impaired growth markedly, with the strength of impairment roughly correlating with an increase in the time of residence of F-actin in patches and a decrease in the endocytic uptake of cargo, emphasizing the master role of endocytosis in the filamentous fungal mode of growth. Notably, our experiments strongly support the conclusion that formin does not play any role in endocytosis, and that the source of mother filaments that prime Arp2/3 networks is dependent on Dip1. Lastly, our data indicate that Dip1 cannot be the only source of mother filaments, and suggests that this Dip1-independent mechanism might be mediated by cortical actin filaments. Assisted by these data, we propose a model that explains the coupling between the growing tip and the subapical endocytic collar, which was denoted the tip growth apparatus by Oakley et al. [5].

## Results

### Tropomyosin labels both the Spitzenkörper and the actin cables radiating from it

As actin cannot be functionally labeled with GFP, *in vivo* studies of the actin cytoskeleton rely on imaging fluorescent protein-labeled F-actin interactors which serve as proxies. In filamentous fungi, translational fusions of the F-actin-binding oligopeptide Lifeact to fluorescent proteins (FP) have been used in *N. crassa* [22,27], *A. nidulans* [28] and *Magnaporthe grisea* [29]. Even though FP-Lifeacts display broad specificity, it is well established that certain structures can be visualized only with specific reporters [30,31]. A remarkable example in *A. nidulans* is the mesh of actin cables radiating from the Spitzenkörper (SPK). The SPK is an accumulation of secretory vesicles localizing underneath the apical plasma membrane that acts as a vesicle supply center [32,33]. These cables serve as tracks for the myosin-5-dependent delivery of secretory vesicles to the apex [34–37]. Exocytic actin cables have been documented using immunofluorescence microscopy of actin in *A. nidulans* fixed cells [6,38]. However, filming these cables *in vivo* has proven elusive. The standard reporter used to visualize them is a GFP-tagged version of tropomyosin [39], a protein essential for growth (S1 Fig) that plays a key role in the efficacy of myosin-5-dependent transport [5,40–42]. Previous attempts to label tropomyosin endogenously with GFP suggested that N- and C-terminally GFP-tagged versions of tropomyosin expressed at physiological levels were not fully functional [5]. We used the heterokaryon rescue technique [43] to establish that a *tpmA-gfp* allele generated by endogenous tagging is indeed unable to sustain viability on its own (S1 Fig), indicating that the presence of GFP impairs TpmA function. Therefore, using TpmA-GFP to illuminate exocytic cables requires co-expression of the untagged protein [5].

Recently, the dynamics of tropomyosin cables has been studied after driving expression, with the *alcA* promoter, of N-terminally tagged GFP-TpmA, in a strain co-expressing the untagged protein to sustain viability. A limitation of this

study was that switching on this promoter required utilization of 2% (~0.22 M) glycerol as carbon source [44]. As we were interested in filming TpmA under more physiological conditions, we set out to develop alternative reporters based on FP-TpmA that could provide a sufficiently bright signal of the fusion protein in the presence of the necessary backup copy of the untagged version. Firstly, we extended the linker sequence between the tropomyosin C-terminus and the FP from fifteen to eighty residues and used mNeonGreen instead of GFP, as successfully done in *Schizosaccharomyces pombe* [45]. However, this modification resulted in a largely cytosolic localization. Secondly, we constructed a transgene encoding a C-terminal fusion of TpmA to tdTomato (tdT), driven by the *tpmA* promoter. This, and its twin GFP-tagged construct were targeted in single copy to the *white* locus. These two C-terminally tagged TpmAs labeled the mesh of actin cables at the tip, but TpmA-tdT was brighter. Fig 1A shows that TpmA-tdT reveals a 'mop' of actin cables radiating from the SPK that extend basipetally 2–3 µm away from the latter. Remarkably, all reported tropomyosin constructs including ours illuminated the SPK, which strongly indicates that this structure contains a mesh of actin filaments whose organization is similar to that of filaments used for myosin-5 transport.

### Lifeact labels actin in the SPK, endocytic patches, septa and dynamic actin filaments

Lifeact could modify the dynamics of actin when expressed at high levels. For example, it binds to an epitope on F-actin that overlaps that of the actin filament-severing protein cofilin [46], potentially stabilizing actin filaments. To prevent unwanted effects, we took advantage of the promoter of the *inuA* gene encoding *A. nidulans* inulinase [47,48]. This gene is inducible by sucrose in a concentration-dependent manner and, unlike most genes required to catabolize alternative carbon sources, is not repressible by glucose, which behaves as a neutral carbon source resulting in low basal levels of expression. By adjusting the carbon source in the culture medium, we were able to tune up expression levels of FP-Lifeact constructs. For example, steady-state levels of Lifeact-GFP obtained on a mixture of 0.05% glucose and 0.025% sucrose were 10 times higher than those on 1% glucose (Fig 1B). Expression levels of Lifeact on 0.05% glucose and 0.025% sucrose resulted in a bright GFP signal and did not affect apical extension rate (Fig 1C and S1 Movie), which is highly sensitive to perturbations of the actin cytoskeleton. Thus, these conditions were routinely used.

GFP and tandem dimer tomato (tdT) versions of Lifeact, tagGFP- and mChFP-tagged versions of an anti-actin nanobody, GFP-actin and GFP-tractin labelled four different structures, namely contractile actin rings (CARs) at sites of septation, endocytic actin patches, actin cables, and a bright apical spot at the position of the SPK (Fig 1D). That this spot indeed corresponds to the SPK was initially confirmed by its labeling with FM4–64 (Fig 1E). However, we note that unlike the case of other filamentous fungi, such as *N. crassa*, FM4–64 labels the *A. nidulans* SPK only very faintly [49], and does so in a small number of tips [23,50]. Moreover, it labels very strongly the mitochondria [23,49], making even more difficult the reliable detection of the SPK with this dye. Thus, we determined independently that the apical bright spot of F-actin is the SPK by its colocalization with the type V myosin MyoE (Fig 1F), a well-established marker of this structure [36,51].

With the obvious exception of CARs, and as reported by previous immunofluorescence studies [6,38,52], the actin cytoskeleton revealed by the above FPs is strongly polarized (Fig 1D and S1 Movie), mostly as a result of the high abundance of endocytic patches in the subapical collar [4–6]. In *A. nidulans* this collar is located, on average, 1.3 µm away from the hyphal apex and, in two-dimensional projections, defines the base of the apical dome semi-circumference, whose perimeter is, approximately, 4.5 µm (Fig 1H and 1I). Fluorescence intensity indicated that endocytic patches are at least three times more abundant per surface unit in the collar than in regions more distal to the tip (Fig 1G). Time-lapse sequences illustrated the coordinated advance of the endocytic collar and the SPK as hyphae grow by apical extension (Fig 1K).

Co-filming of β-tubulin-GFP with tdT-Lifeact revealed that in our diffraction-limited images the plus ends of microtubules (MTs) converge at the apical region, at or near the position of the SPK (Fig 2A), confirming previous reports on the organization of the MT cytoskeleton at the hyphal tips [50,53,54]. Fig 2B shows an example of a MT reaching the SPK, and S2A Fig shows a time-lapse sequence containing several instances of MTs converging at the apical region, both at the SPK itself as well as in its immediate vicinity. Fig 2C and S2 Movie show that the organization of the actin cytoskeleton persists

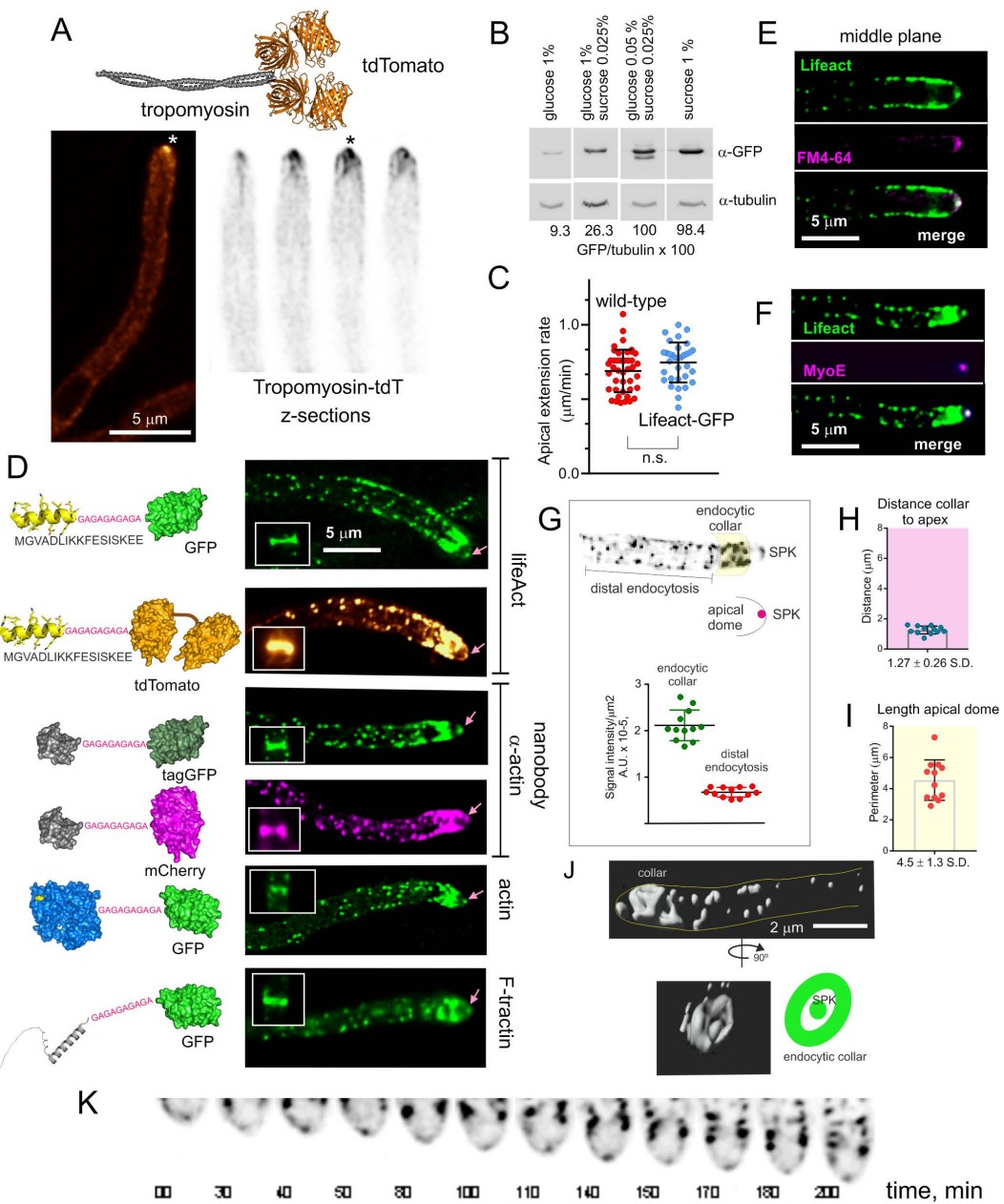

**Fig 1. A summary of the reporters used to detect F-actin structures. (A)** Localization of TpmA-tdTomato (tdT). Top, ribbon scheme of TpmA-tdT. Bottom, left: The image is a deconvolved middle plane of a hyphal tip cell showing the localization of TpmA-tdT to the SPK (asterisk). Bottom, right: consecutive planes of a Z-stack (Z = 0.5 μm) showing the SPK (asterisk) and the actin cables that emerge from it. **(B)** Western blot analysis of Lifeact-GFP expression levels on different carbon sources. Tubulin was used as loading control. **(C)** Expression of Lifeact-GFP does not cause defects in apical extension. Quantitation of apical extension rates in the wild-type and in a Lifeact-GFP strain. Hyphae were cultured on WMM containing 0.05% (w/v) glucose and 0.025% (w/v) sucrose as carbon sources. Time stacks acquired with Nomarski optics at 28°C were used to calculate apical extension rates of individual cells with kymographs. Error bars indicate the mean ± S.D. Means were compared by Student's *t*-test; ns, non-significant. **(D)** Localization of the different actin reporters. Left, schematics depicting the structures of the different actin reporter constructs. Right, fluorescence images of the localization of the different actin reporters. Images are maximal intensity projections (MIPs) of deconvolved Z-stacks. The SPK, endocytic collar, actin cables, contractile actin rings (CARs, shown in boxes) and actin patches can be observed. **(E)** The apical spot of F-actin colocalizes with the SPK, labeled with FM 4-64, which marks SVs corresponding of the endocytic recycling pathway. The images are middle planes of a deconvolved Z-stack. **(F)** The apical spot of F-actin colocalizes with the SPK resident MyoE tagged with mChFP. Images are MIPs of deconvolved Z-stacks. **(G)** The density of actin patches in the endocytic collar is at least three times higher than in distal regions of the hypha. Lifeact-GFP image (middle plane) showing SPK, actin patches

and endocytic collar (shadowed yellow). Graph quantifying the signal intensity of Lifeact-GFP in the endocytic collar region and in distal regions of the hypha. Error bars show mean ± S.D. **(H)** Graph representing the distance from the endocytic collar to the apex. Error bars show mean ± S.D. (*N* = 12). **(I)** Graph representing the perimeter of the apical dome. Error bars show mean ± S.D. (*N* = 12). **(J)** Three-dimensional reconstruction of a confocal z-stack showing the spatial relationship between the endocytic collar and the SPK. **(K)** Time-lapse sequence of the F-actin cytoskeleton at the tip visualized with Lifeact-GFP. The Movie illustrates how the endocytic collar and the SPK advance coordinately.

during mitosis, reflecting the fact that despite that most MTs are depolymerized during the nuclear division cycle to supply tubulin monomers for the mitotic spindle, a subset of cytoplasmic MTs remain stable and suffice to maintain normal rates of apical extension [55] and, as shown by to this experiment, the integrity of the actin cytoskeleton.

Takeshita et al. have proposed that MTs are guided to the apex by the type V myosin MyoE moving on actin cables [53], implying that the complete depolymerization of F-actin should disperse across the apical dome the sites at which the plus-ends of MTs contact the cortex. Latrunculin B at 50 µM completely depolymerizes F-actin within 5 min [9,56]. Indeed, left panels on Fig 2D show how seven minutes after adding the drug these contact sites were dispersed across an apical dome. This has undergone a characteristic swelling because tip growth continued, fueled by secretory vesicles transported by the MTs, but it was no longer polarized towards the apex. That this 'semi-isotropic' growth continues without F-actin is illustrated by the much larger dimensions of the swollen tip depicted on the right Fig 2D panels, corresponding to 60 min of incubation. Even though MTs are disorganized and bent at this time point, some of them reach the cortex (S2B Fig). In short, our data add to previous reports showing the actin-dependence of microtubule organization. Also as reported previously [5],The complete depolymerization of microtubules with benomyl (5 µM) resulted in an almost complete disorganization of the endocytic actin collar and the absence of SPK (Fig 2E), correlating with bulging of the tips [5]. Upon washing out the drug, hyphal tips grew slowly for a few minutes (0.042 µm/min in the example shown on S3 Movie) (details in S2C Fig), and gradually resumed fast hyphal tip growth (to 0.56 µm/min) as the actin collar and the normal tip morphology were recovered and a new SPK became patent S2C Fig). Therefore, these data further illustrate the microtubule-dependence of the actin cytoskeleton. This mutual dependence between actin and microtubule cytoskeletons very likely mediates the intimate spatial relationship between the endocytic collar and the growing apex, denoted "the tip growth apparatus" by Oakley and collaborators [5].

### Fluorescent protein-tagged actin, anti-actin nanobody and tractin behave as Lifeact

We wondered if the inability of Lifeact to reveal tropomyosin-containing cables is specific for this reporter as would occur if, for example, actin-binding regulatory proteins compete with it for an overlapping binding site. Wee designed an alternative F-actin reporter consisting of a high affinity anti-actin nanobody (chromobody from ChromoTek) [30] fused to either tagGFP2 or mChFP, also expressed under the control of the inulinase promoter. Similar to Lifeact, these reporters did not label actin cables radiating from the apex (Fig 1D). In a further attempt to detect tip cables, we constructed a GFP fusion protein with F-tractin [31], which behaved essentially as Lifeact and nanobody actin reporters (Fig 1D). These data were compared with those obtained with actin, N-terminally tagged with GFP. As numerous reports have concluded that FP-tagged versions of actin are not fully functional, we expressed GFP-actin under the control of the inulinase promoter in a strain backed by a resident copy of the actin gene. GFP-actin was detectable in actin patches, with conspicuous labelling of the endocytic collar, and in the SPK, but it did not label actin cables (Fig 1D and S4 Movie). There are several possibilities that might explain the inability of all the above probes to detect actin cables. One is that cable-specific regulatory factors such as tropomyosin efficiently compete for binding with Lifeact. Another, that the high affinity actin binding domains (Lifeact, chromobody, tractin) recognize G-actin and/or nascent filaments too efficiently, causing steric hindrance with formin and preventing their incorporation into actin cables [30]. However, neither of these possibilities would explain why "exocytic" actin cables are clearly visualized with Lifeact in *M. oryzae* [29]. Therefore, we speculate that the high concentration of actin patches containing branched actin in the tip might overcompete cables with regard to the availability of

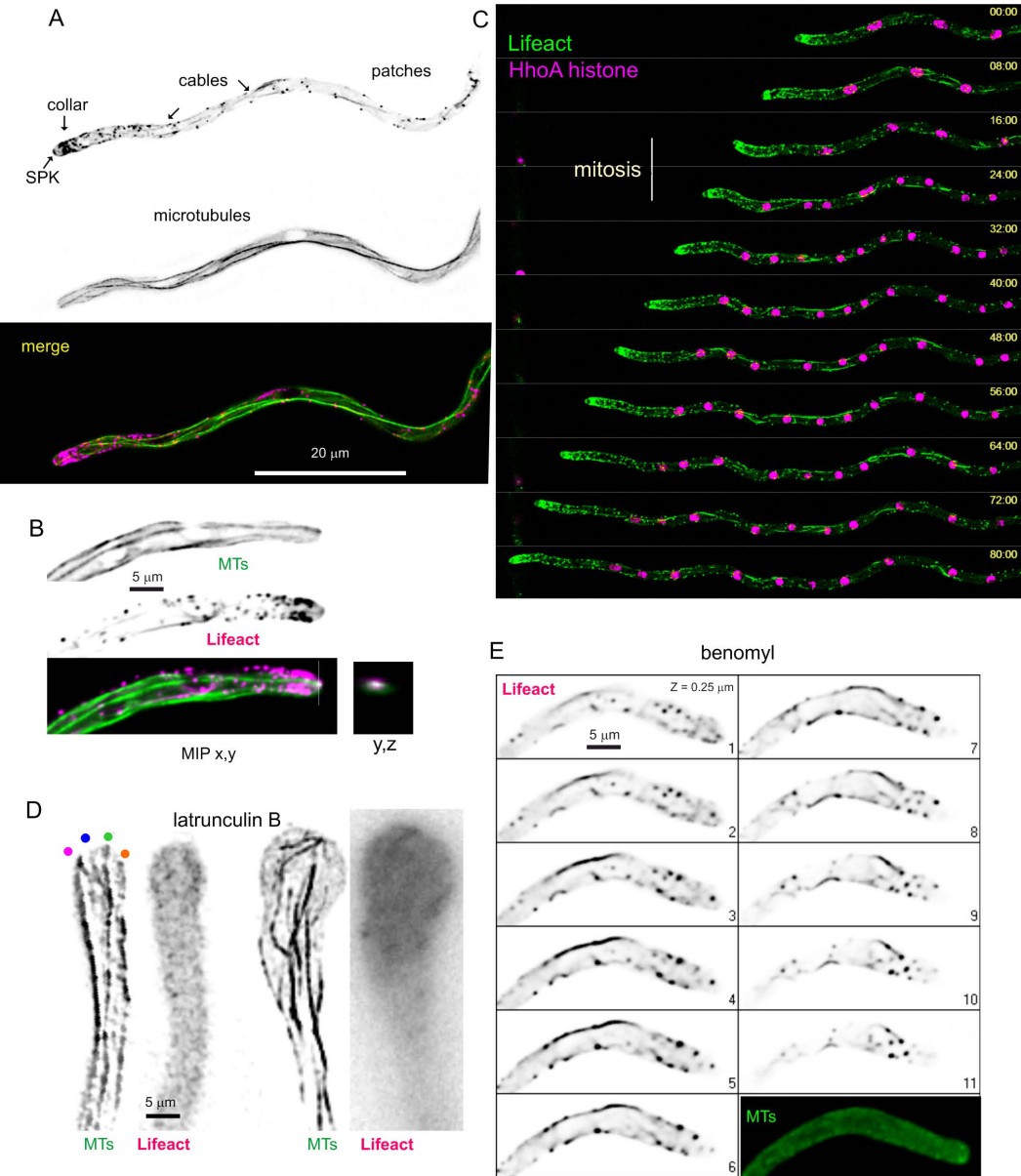

**Fig 2. The actin cytoskeleton visualized with Lifeact. (A)** Relationship between actin and microtubule cytoskeletons. F-actin visualized with Lifeact-tdT and MTs with TubA-GFP. Images are MIPs of deconvolved Z-stacks. **(B)** Hyphal tip cell illustrating how the plus-end of one MT reaches the SPK. Images are MIPs of a dual-channel Z-stack of F-actin labeled with Lifeact-tdT and MTs labeled with TubA-GFP. The box on the right is a (y,z) orthogonal view of the region of the SPK, showing colocalization. **(C)** Different frames of a time-lapse sequence (S2 Movie) showing F-actin labeled with Lifeact-GFP and chromatin labeled with HhoA-mChFP. **(D)** Disorganization of the MT cytoskeleton upon F-actin depolymerization with 50 μM latrunculin B. Left images correspond to a hypha that has been treated with latrunculin B for 7 minutes. Four microtubules contact the apical dome at distant sites from each other (indicated with colored circles). The image corresponds to a middle plane of a deconvolved Z-stack that has been treated with the unsharp mask filter of Metamorph. Right images correspond to a hypha that had been treated for 70 minutes with latrunculin B. Note the remarkable swelling of the tip and the complete disorganization of the MT cytoskeleton. MTs were visualized with GFP-tagged alpha-tubulin, whereas actin was detected with Lifeact-tdT. MIP of a deconvolved Z-stack. **(E)** The organization of the F-actin cytoskeleton is dependent on microtubules. Different focal planes of a Z-stack of a hyphal tip cell expressing Lifeact-tdT to visualize actin and alpha tubulin-GFP to visualize the microtubule cytoskeleton. Cells had been treated with benomyl at 5 μM. Note that while MTs are completely depolymerized, F-actin is not, although the F-actin cytoskeleton is completely disorganized and the actin collar is no longer present.

the reporters, an interpretation consistent with data presented below. In any case, these experiments confirmed, using five different probes *in vivo*, the presence of F-actin in the *A. nidulans* SPK, a fact which had been thoroughly reported previously, and armed us with a set of bright fluorescent fusion proteins that appeared to be well suited to study the dynamics of F-actin in endocytic patches.

## Analysis of endocytic patches

In view that all the above-tested reporters gave similar results, efficiently labeling endocytic actin patches, and that Lifeact was the smallest actin-binding epitope, in all experiments described below we used the two fluorescent constructs of Lifeact detailed in the preceding paragraph as reporters for F-actin in endocytic patches. These were sufficiently bright to permit the acquisition of several minute-long time series with 1–10 fps time resolution, (S5 Movie is Lifeact-GFP at 5 fps). These long periods of illumination did not arrest rapid apical extension, as shown by the tilted lines drawn by the endocytic collar in longitudinal kymographs using linear ROIs traced across the long axis of the hypha and covering its whole width (Fig 3A, this hypha was growing at 1.2 μm/min and filmed at 2 fps). In these plots, the actin-containing phase of each endocytic event was visualized by a more or less vertical line, whose height reflects the time of residence of F-actin in these structures. Even at the 2 fps time resolution used in this experiment, this kymograph clearly illustrates the remarkable number of endocytic events that take place per unit time and reveals that actin patches have static and motile phases, in that order.

We determined that the time resolution that was optimal to analyze these events was 5 fps, which was the standard resolution used throughout this work, unless otherwise indicated. Fig 3B shows a typical longitudinal kymograph for a 5 fps Lifeact-GFP time series. The most abundant pattern of actin patch behavior is shown on Fig 3C, left patches, both for GFP- and tdT Lifeact examples, with Fig 3D showing a frame-by-frame montage of another example. In these patches, actin arrived at cortical structures that remained static for about two thirds of their lifecycle and subsequently underwent short range basipetal and acropetal movements before disappearing. The static phase is reflected by a straight vertical line whereas the motile phase gives rise to a wiggly trace. The average residence lifetime of F-actin in these patches was 13.7 sec and 13.4 sec with the GFP and the tdT Lifeact reporters, respectively (Fig 3E). This pattern was compared to that obtained with transversal ROIs, perpendicular to the axis of polarity. These kymographs, constructed with middle planes, showed that F-actin remained cortical during most of the patch lifetime, a point at which it appeared to move inwardly and subsequently disappeared (Fig 3C, right patches, several examples displayed in S3B Fig). The lifetime of patches using transverse kymographs was 13 sec, which is consistent with the values obtained with longitudinal ones. As patches move in both directions during the wiggly phase and, in addition, move forward with the cell which is growing rapidly, capturing the whole lifecycle of a patch using a transversal ROI requires a wide line, implying that the chances of another patch entering into the ROI are very high. This limited the number of events that could be analyzed. In contrast, longitudinal kymographs drawn will linear ROIs covering at least half of the hyphal width, facilitated the detection of a high number of patch lifecycles per time sequence. However, given that it was often difficult to identify individual patches whose traces were not overlapping, we converted x, y, t stacks into x, y, z stacks that could be analyzed with a 3-D viewer, which facilitated the neat separation of traces that overlapped in 2D (S3A Fig and S6 Movie). Remarkably similar times were measured with chromobody-GFP, F-tractin-GFP and actin GFP, and only a slightly shorter one (11.9 sec) with chromobody-mChFP (Fig 3E) (the later minor discrepancy attributable to the highest sensitivity of mChFP to photobleaching).

We acknowledge that we took the point at which the wiggly lines are no longer visible as the point at which the patch transits to invagination, but we have not identified formally this point. The reproducibility of the patch lifecycles, the consistent values obtained with longitudinal and transverse kymographs and the coincidence in the duration of the lifecycles among different components of the actin phase of endocytosis strongly support the validity of this assumption.

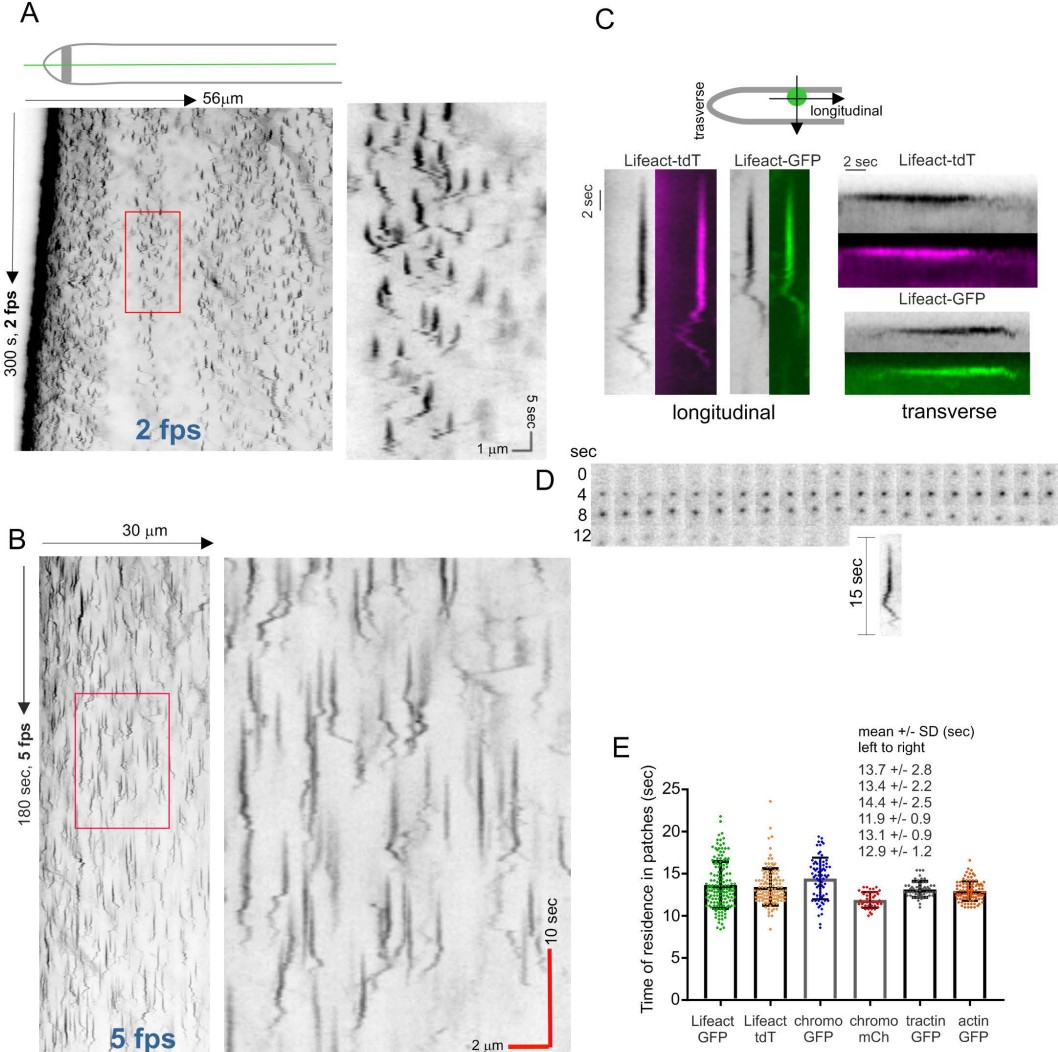

**Fig 3. Determining the behavior and lifetime of actin patches visualized with F-actin reporters. (A)** Kymograph derived from a time-lapse sequence of Lifeact-GFP acquired with a time resolution of 2 fps. The ROI was a line traced longitudinally across the hyphae whose width covered the whole hypha (see scheme on top). The indicated rectangular region is magnified on the right. Note the tilted line corresponding to the endocytic collar is moving ahead with tip growth, demonstrating that imaging conditions do not impair growth. **(B)** Kymograph derived from the time-lapse sequence of Lifeact-GFP made with images acquired every 0.2 seconds for 3 minutes. The indicated rectangular region is magnified on the right. **(C)** Behavior of individual actin patches. Images obtained from longitudinal (left) or transverse (right) kymographs. **(D)** Montage showing the progression of one Lifeact-GFP actin patch, whose kymograph trace is shown below. **(E)** Half-life, in seconds, of actin patches labeled with the different reporters indicated in the graph. Error bars indicate mean ± S.D. Values were obtained from individual patches extracted from kymographs with a time resolution of 5 fps.

## Actin patches associate with cables and often appear to nucleate them

Z-stack series sharpened with deconvolution algorithms clearly showed actin patches associating with cables (Fig 4A and 4B). While estimating the degree of proximity between actin patches and cables is diffraction-limited, two cases strongly support the contention that patches and cables are intimately associated. First, association between patches and cables is observed in essentially every cell. Second, in time-series, patches and cables remain associated across the whole lifetime of the patch, which is difficult to explain if the two structures overlapped by chance (Fig 4C and S7 Movie, focused

on a long cable). In addition to patches progressing closely associated to cables, we recorded examples in which cables appear to be nucleated from patches (Fig 4D). In fact, faint filamentous structures connecting adjacent patches and undergoing cycles of growth and shrinkage can be observed by increasing contrast of the images. These faint filaments eventually appear to nucleate a neighboring patch (S8 Movie). Assuming the correctness of the dendritic nucleation model, we hypothesize that these connections might correspond to filaments that have not yet been severed by cofilin.

3D (x,y,t) and 4D (x,y,z,t) time series revealed that large pieces of actin cables are being continually drained away from the tip towards basal regions of the hyphae. These chunks, which we denoted "actin worms" for its characteristic crawling-like way of movement (S9 Movie), result from fragmentation of long actin cables connecting large rows of patches (S10 Movie, see how the long cable at the bottom edge of the hypha is fragmented into "worms"). Further examples of such phenomenology can be clearly seen in the series displayed on Fig 4E, in which a long cable is severed in the middle, the tip-proximal part is depolymerized (shaded in blue) and the tip-distal part moves away from the tip while simultaneously shrinks by its trailing end (shaded in yellow). As this occurs, a new cable (shaded in magenta) grows by its leading end before shrinking by the trailing end. The movement of these actin "actin worms" was captured by kymographs, in which they give rise to diagonal lines reflecting their movement away from the tip (Fig 4F). These lines are blurred because the intensity of the signal across actin cables is not uniform. In the example displayed on Fig 4F, corresponding to a time resolution of 2 fps, several of these pieces depart from the endocytic collar, and the parallel diagonal lines to which they give rise in kymographs indicate that they move at approximately 0.4 µm/s. The examples shown on Fig 4G (an x,y,z,t 4D-series corresponding to S11 Movie) illustrate that these movements can be considerably long. This movie captures one of these actin cable fragments moving towards the tip, which is exceptional, as in a vast majority of cases they move away from the tip. It also illustrates that length of these runs can be quite considerable. The mechanism propelling these "actin worms" is currently unknown (see below), but we note that if these "actin worms" moved away from the tip by treadmilling, the leading end of these cables should be the barbed end; which is not compatible with the polarity expected if these would have been originated directly by severance of longer filaments nucleated in the apical cortex.

The above experiments did not give any clue on the relationship, if any, between these patch-associated "worms" and the mesh of tropomyosin-containing cables generated by formin at the SPK, an issue that will be addressed below. Only in exceptional time series like that shown on Fig 5 and S12 Movie, a connection between a nascent "worm" and cables derived from the SPK was noticeable. Here, a long cable associated with patches at the base of the tip appears to originate at a diffuse concentration of actin sandwiched between the two sides of the collar, at which a short actin cable emanating from the apex appears to reach (Fig 5-a1). These connections are evident on Fig 5-a2 frames, in which the SPK is visible. Eventually, this long cable is released from the tip and moves basipetally, while a second actin worm is generated at considerable distance from the apex (Fig 5-a3). The exceptionality of such a connection, the intimate association between worms and patches and the fact that worms arise often from basal regions argues against this chunks of actin deriving from cables emanating from the SPK.

## Actin polymerization occurs on SlaB<sup>Sla2</sup> patches

The burst in actin polymerization at cortical patches reported by the above probes represents the final stage of an endocytic pathway akin to that extensively studied in *S. cerevisiae*, of which we used AbpA<sup>Abp1</sup> (*i.e.,* the homologue of the budding yeast Abp1) as a prototypic component. AbpA<sup>Abp1</sup>, tagged endogenously with GFP, localizes to patches concentrating in the subapical endocytic collar (Fig 6A and S13 Movie). In time-lapse series, AbpA<sup>Abp1</sup> showed the same behavior as F-actin in patches. The AbpA<sup>Abp1</sup>-GFP time of residence was 13.2 ± 1.7 S.D. seconds (Fig 6B and 6C), which is basically identical to that of Lifeact reporters, as expected for a protein belonging to the actin polymerization network. AbpA<sup>Abp1</sup> is absent from the SPK, further indicating that F-actin in the SPK differs from that in endocytic patches (but see below).

In the budding yeast Sla2 cooperates with Ent1 to couple the membrane-associated endocytic patch coat containing cargo-loaded adaptors to the actin polymerization machinery [57,58]. The gene encoding *A. nidulans* SlaB<sup>Sla2</sup> is essential

PLOS Genetics

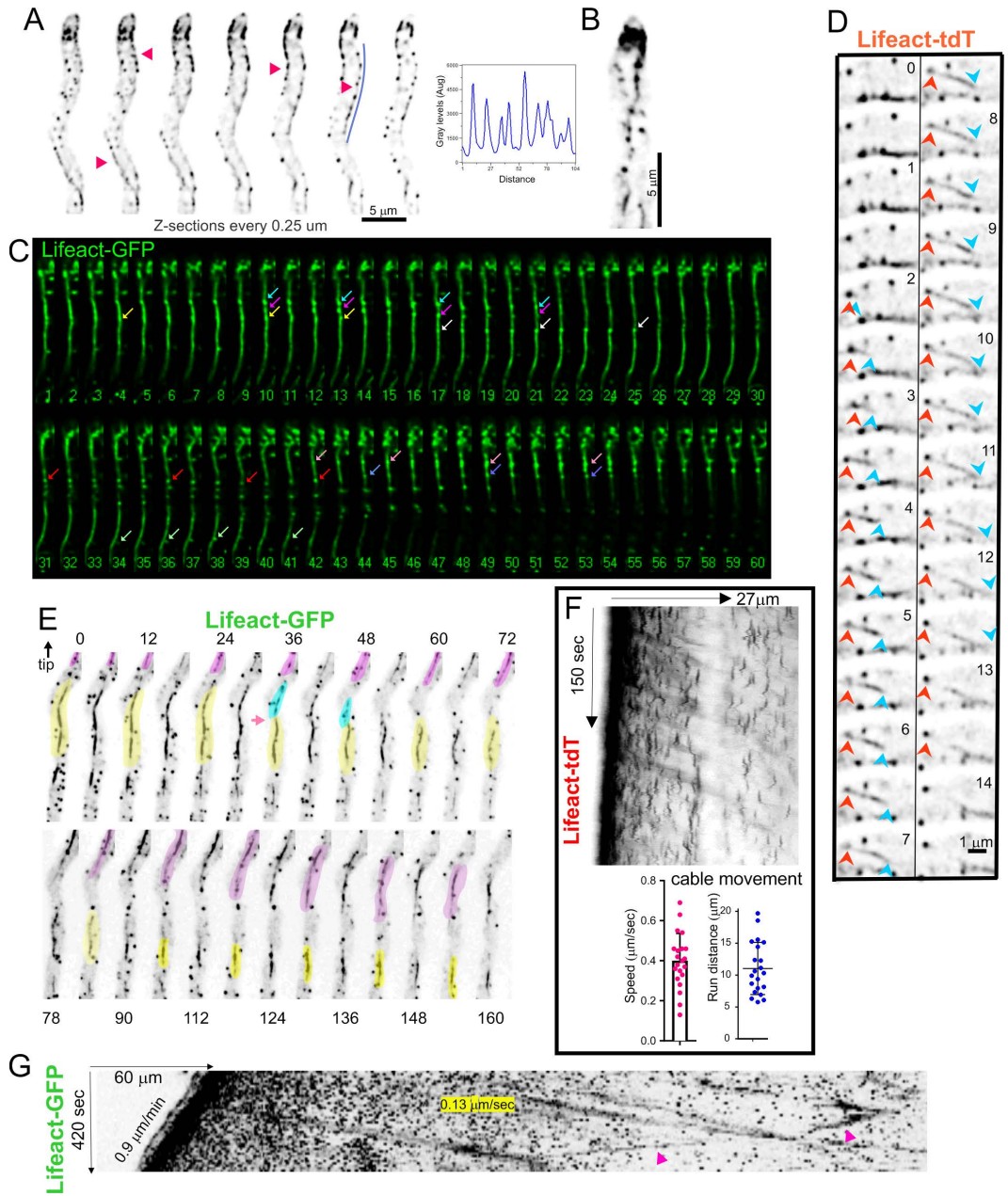

**Fig 4. Actin patches associate with actin cables. (A)** Actin patches are associated with cortical filaments. Consecutive planes of a z-stack of Lifeact-GFP (z = 0. 25 μm). The graph is a linescan across the ROI indicated. **(B)** MIP of a z-stack of Lifeact-GFP. **(C)** Still images extracted from S7 Movie showing the formation of patches associated with cables (arrows, each patch is color-coded). Numbering indicates time in seconds. **(D)** Still images from a time-lapse acquisition at 2 fps showing an example of an actin cable (tail indicated by a blue arrowhead) nucleated from a patch (red arrowhead). **(E)** Still images showing different aspects of the dynamics of actin cables labeled with Lifeact-GFP. Numbers indicate time in seconds. The tip is at the top. The pink-shaded cable grows during the first two minutes of the sequence before moving towards the base. The yellow-shaded cable is severed at the 36 seconds time-point (arrowed). The blue-shaded portion appears to be depolymerized, whereas the yellow portion moves towards the base as it is being depolymerized **(F)** Kymograph derived from a Lifeact-tdT time series acquired at 5 fps. Actin cables moving away from the tip, occupied by a prominent actin collar, are seen as diagonal lines. The graphs represent the speed of this basipetal movements and the average length travelled by these cables. Error bars indicate means S. D. **(G)** This kymograph was derived from a time-lapse sequence in which images were acquired at 6 fpm. The ROI was a 30 pixel wide line that followed the longitudinal axis of the hypha. Individual patches are seen as dots at this time resolution. The front tilted line is the SPK (this hypha is growing at 0.9 μm/m). The highly fluorescent region behind it corresponds to the endocytic collar. Diagonal lines correspond to the movement of fragments of actin cables (see text). The speed of one such fragment is indicated. This kymograph illustrates the long distances that can be covered by moving fragments of actin cables.

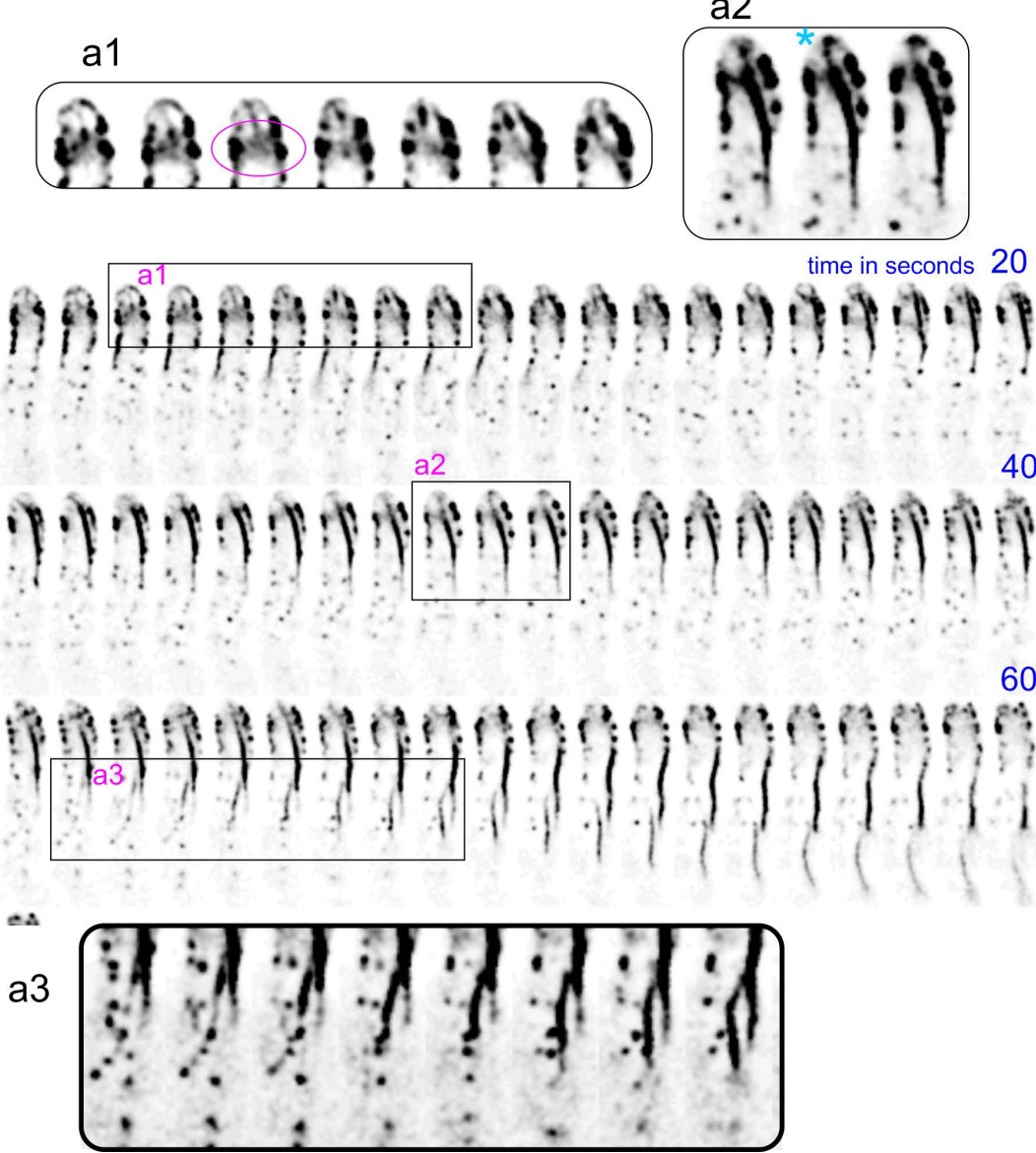

**Fig 5. On the origin of actin worms.** A hyphal tip cell expressing Lifeact-tdT was filmed with a time resolution of 1 fps. In the region magnified in a1 the region located at the base of the apical dome (circled in magenta) appears to receive F-actin both from the endocytic collar and from an actin cable emerging from the SPK (magnified region a2, the cyan asterisk indicates the SPK), which results in the formation of a prominent subapical actin cable that grows basipetally until is severed from tip structures and continues its basipetal movement. A second actin cable originating at patches distal from the tip crosses its trajectory (magnified region a3).

[6,24]. Endogenously tagged SlaB$^{Sla2}$ localizes to patches, predominating in the subapical endocytic collar (Fig 6D). With time-lapse acquisitions at 1 fps we determined that the average time of residence of SlaB$^{Sla2}$ in cortical structures is 51 sec ± 15 S.D., similar to that reported for the budding yeast [12], and notably longer than the time of residence of actin itself (Fig 6E and 6F). To demonstrate coupling between endocytic invaginations and the F-actin polymerization machinery, we co-filmed SlaB$^{Sla2}$-GFP with Lifeact-tdT. Kymographs (Fig 6F) clearly established that the burst in F-actin occurred

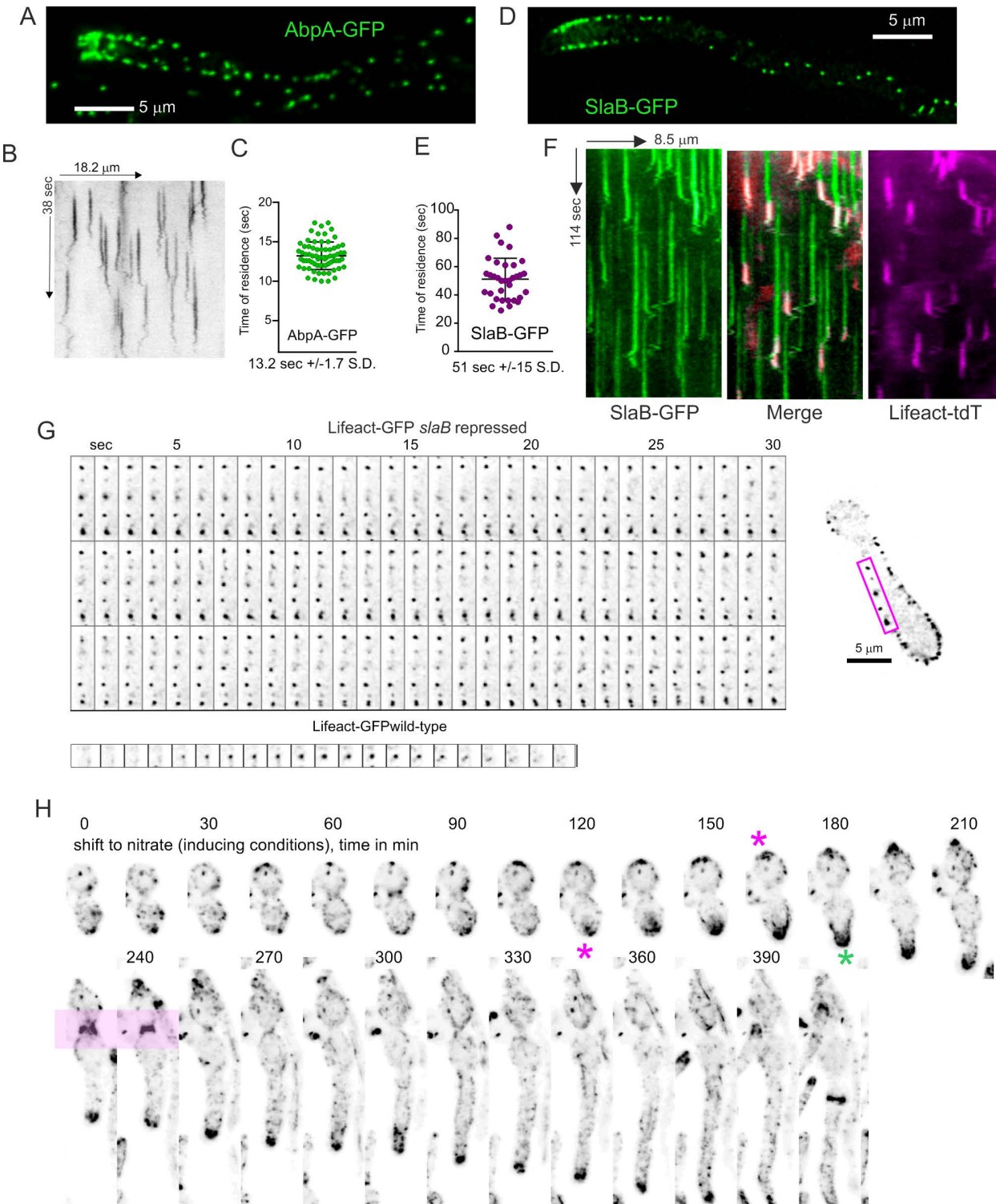

**Fig 6. Localization and dynamics of AbpA[Abp1] and SlaB[Sla2].** **(A)** Localization of endogenously tagged AbpA[Abp1]-GFP. Middle plane of a deconvolved z-stack. **(B)** Kymograph of AbpA[Abp1]-GFP with a time resolution of 5 fps. **(C)** Time of residence of AbpA[Abp1]-GFP in patches. Error bars represent means±S.D. **(D)** Localization of endogenously tagged SlaB[Sla2]-GFP. The image is a middle plane of a deconvolved z-stack illustrating the cortical localization of this protein. **(E)** Time of residence of SlaB[Sla2]-GFP in patches. Error bars represent means±S.D. **(F)** Kymographs derived from a time-lapse sequence with simultaneous acquisition of Lifeact-tdT and SlaB[Sla2]-GFP channels. **(G)** Montage constructed with individual frames of S14 Movie

corresponding to the abortive germling shown on the right, which expresses Lifeact-GFP. This is strain carries a conditional allele permitting expression of SlaB[Sla2] on nitrate- but not on ammonium-containing medium. Conidiospores were germinated on ammonium, and therefore cells are deficient for SlaB[Sla2]. The resulting germlings were able to establish polarity but not to maintain it, arresting growth with a characteristic swollen tip typical of a defect in polarity maintenance. In these germlings, Lifeact-GFP appears as clumps associated with the plasma membrane. The montage at the bottom correspond to a wild-type strain germinated under the same conditions, with individual frames depicted with the same time resolution to facilitate comparison. **(H)** Conidiospores carrying the ammonium-repressible and nitrate-inducible *slaB* allele indicated above were germinated on ammonium before being shifted to nitrate medium to induce the synthesis of SlaB[Sla2]. At about two hours after the shift F-actin started to concentrate in one of the poles and polarity was reestablished, which resulted in hyphal tips with normal morphology that underwent rapid tip growth. Note the two septation events that take place during these morphogenetic transitions, one of which is shaded in pink. Magenta asterisks indicate the sites of F-actin repolarization whereas a green asterisk indicates the emergence of a morphologically normal tip in which actin cables were noticeable.

at the end of the SlaB[Sla2] cycle, recapitulating the situation in *S. cerevisiae* [12]. Of note, the total time of the cargo selection phase plus the actin polymerization phase is approximately 1 min, like in the budding yeast.

Although *slaBΔ* is lethal, a conditional allele driving expression of the protein under the control of nitrate-inducible, ammonium-repressible promoter is available [24]. Conidiospores germinated on ammonium (i.e., without SlaB[Sla2] expression) arrested growth shortly after establishing polarity, with a characteristic morphology shared by null mutants in every essential gene of the *A. nidulans* endocytic internalization pathway. In these abortive germlings, F-actin appears as characteristic clumps hanging from the cortex (S14 Movie). These actin patches are not disassembled (Fig 6G) and very likely represent unproductive events of endocytosis due to the uncoupling between the clathrin module and the F-actin polymerization module (Fig 6G, frames corresponding to a wild-type control are shown for comparison below those of the SlaB-deficient patches). As a further case supporting the conclusion that the polarization of the actin collar is required for polarity maintenance, we shifted cells cultured overnight on ammonium, arrested with the characteristic phenotype of cells deficient in endocytosis, to nitrate-containing medium. After approximately two hours after the shift, actin patches started to polarize towards the apex and the endocytic collar was gradually being recovered, correlating with resumption of growth and the transition towards a normal hyphal tip morphology and rapid apical extension (Fig 6H and S15 Movie). Thus, these data are consistent with the existence of different protein modules that act sequentially during endocytic internalization, which have been thoroughly studied in the budding yeast.

To buttress this conclusion, we analyzed another component of the coat complex acting upstream of the actin polymerization stage [12], the EH (epsin homology) domain-containing protein End3, which has been previously denoted SagA in *Aspergillus nidulans*. As reported [59], endogenously tagged SagA[End3]-GFP localizes to patches that predominate in a subapical collar that corresponds, as indicated by its colocalization with Lifeact-tdT, to the endocytic collar (S4A Fig). Kymograph analysis of time-lapse acquisitions at 0.5-1 fps showed that SagA[End3] patches lasted for 47.7 sec ± 10.4 S.D. (S4B and S4C Fig) and that their lifecycle preceded that of Lifeact (S4D and S4E Fig). Taking together, all the above data corroborate the existence of two sequentially-acting complexes corresponding to the coat complex (SlaB[Sla2] and SagA[End3]) and the actin polymerization complex (AbpA[Abp1], and FimA and ArpC1, see below) described in *S. cerevisiae* [12].

## Fimbrin is an essential protein present in actin patches and absent from exocytic structures

Another key player in endocytosis is the actin bundling protein fimbrin. In the budding yeast, fimbrin (denoted Sac6) cross-links actin, which appears to be important to generate the force that powers endocytic internalization [12,60]. Previous work with *A. nidulans* AbpA[Abp1] tagged endogenously with FPs [5,6], and with GFP-tagged FimA (*A. nidulans* fimbrin) expressed under the control of a heterologous promoter [4], was instrumental to establish the presence of the characteristic subapical endocytic collar. We corroborated these observations using endogenously GFP-tagged FimA. As reported [4], fimbrin-GFP is present in patches that are markedly concentrated in the subapical collar. Notably, these experiments confirmed that, unlike actin, fimbrin-GFP is absent from the SPK (Fig 7A). Fig 7B illustrates the behavior of FimA-GFP in these patches, with a time of residence of 13 ± 1.07 sec S.D. which matches that observed with actin reporters described on Fig 3.

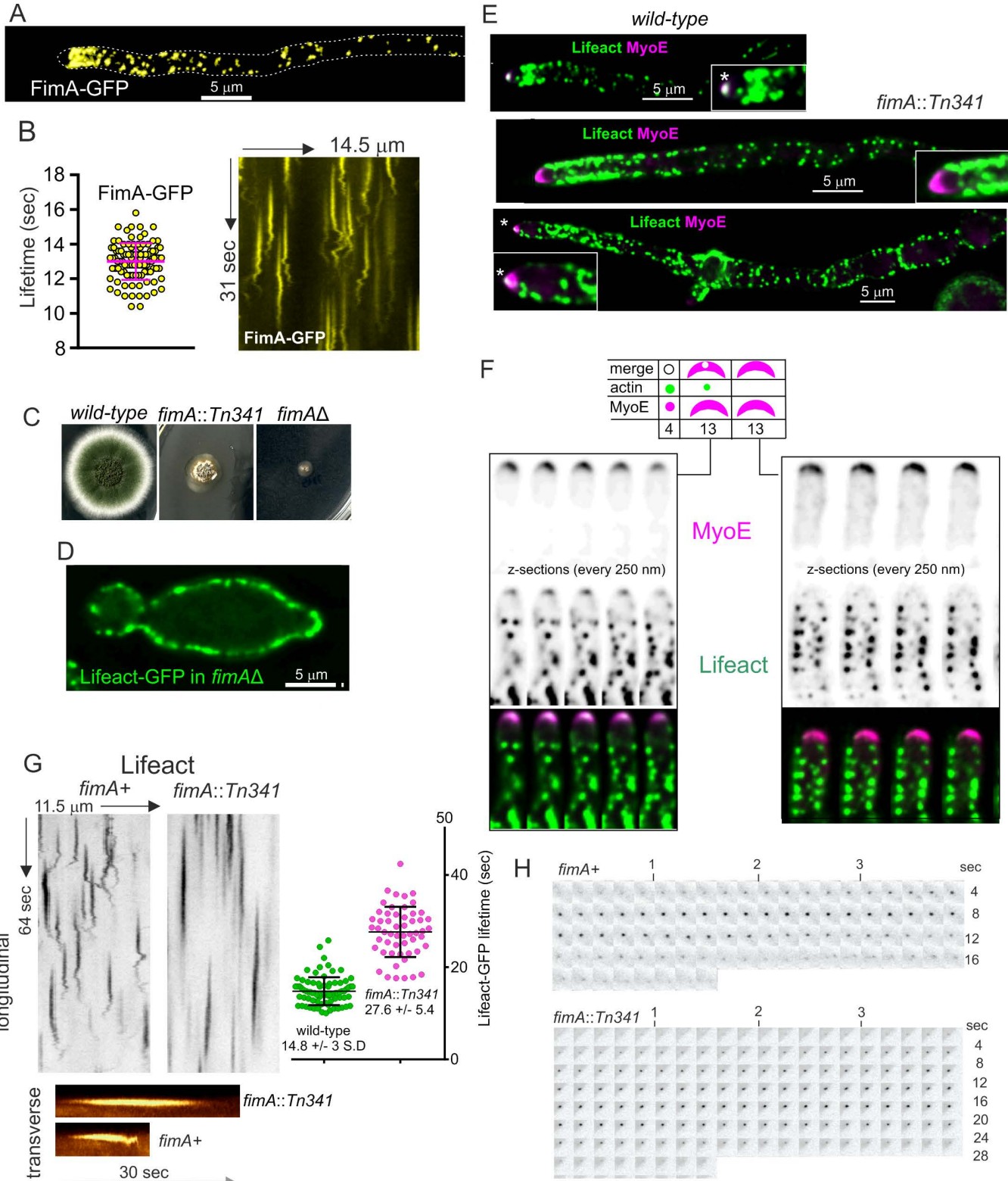

**Fig 7. Functional analysis and dynamics of fimbrin-GFP. (A)** Localization of fimbrin-GFP. The image is a maximum intensity projection of decon-volved Z-stack, pseudo-colored in yellow to distinguish this reporter from images of Lifeact-GFP also included in the Fig. **(B)** Time of residence of

FimA-GFP patches. Error bars represent mean ± S.D. The kymographs were extracted from a Movie acquired at 5 fps. **(C)** Growth test of indicated strains on complete medium at 37°C. **(D)** Terminal phenotype of a *fimAΔ* germling expressing Lifeact-GFP. Spores from a *fimA+ / fimAΔ* heterokaryon were inoculated on selective medium, in which only the *fimAΔ* haploid spores germinate. The image is a MIP of a deconvolved Z-stack. **(E)** Localization of Lifeact-GFP in wild-type and *fimA::Tn341* hyphal tip cells. Images are MIPs of deconvolved Z-stacks. Note the abnormal shape of the SPK monitored with MyoE-mChFP. **(F)** Analysis of the SPK *in fimA::Tn341* hyphal tips. The SPK was monitored with MyoE and then examined for the presence or absence of an apical F-actin-containing spot. 26 out of 30 tips analyzed had an abnormal crescent-shaped MyoE SPK. Of these, 13 showed a weak remnant of apical F-actin in the pole of this crescent. The images are Z- series of deconvolved Z- stacks. **(G)** Dynamics of actin patches in *fimA::Tn341*. Grayscale longitudinal kymographs comparing the behavior of Lifeact-GFP actin patches in the mutant with the wild type. F-actin stays in the patches for a longer period in *fimA::Tn341* and the mutant patches are static. The two data sets were significantly different according to a *t*-student test. The colored traces shown below correspond to transverse kymographs. **(H)** Examples depicting individual frames of wild type and *fimA::Tn341* time series.

*S. cerevisiae* fimbrin, encoded by *SAC6*, plays its key role in endocytic patches by cross-linking F-actin filaments of the dendritic network [60]. Somewhat unexpectedly, an *A. nidulans* disruption mutant of *fimA*, *fimA::Tn341*, in which the open reading frame had been interrupted by transposon mutagenesis, is viable [4], even though it formed morphologically abnormal hyphae and showed delayed polarity establishment. Given that ablation of key endocytic proteins such as Rvs167, SlaB^Sla2, components of the Arp2/3 complex or cofilin is lethal ([6,24] and this work), we hypothesized that *fimA::Tn341* was a partial loss-of-function, rather than a null, allele. The growth test on Fig 7C shows that whereas a *fimA::Tn341* strain is severely debilitated but able to grow to a certain extent, a strain carrying a complete deletion allele of *fimA* is unable to grow, strongly indicating that *fimA::Tn341* is not a null allele. To confirm this conclusion we studied the terminal phenotype of *fimAΔ* using the heterokaryon rescue technique [43]. We constructed a heterokaryotic *fimAΔ/fimA+* strain by transformation, from which we obtained homokaryotic *fimAΔ* conidiospores. These conidiospores were able to establish polarity and to maintain it for a limited time before arresting growth with a characteristic shape that is shared with mutants completely deficient in endocytosis (Fig 7D). In these abortive germlings, F-actin accumulated in bright spots at the periphery that did not progress any further (S16 Movie). Thus, the complete absence of fimbrin abolishes F-actin dynamics to an extent that is incompatible with viability, and *fimA::Tn341* is a partial loss-of-function allele.

We exploited this allele to determine the changes in the actin cytoskeleton that take place when *fimA* is compromised. As reported by Upadhyay et al. [4], *fimA::Tn341* delayed polarity establishment markedly and a large proportion of cells displayed severe morphogenetic defects (S5 Fig). Nevertheless, these abnormal germlings eventually stabilized a polarity axis and gave rise to hyphae. In the wild type, the synaptobrevin-like R-SNARE SynA is polarized by endocytic recycling, localizing to the hyphal tips (S5A Fig). However, in *fimA::Tn341* cells, irrespective of whether they showed abnormal polarity or were relatively normal, SynA localized evenly across the periphery of the cells, indicating that endocytosis was severely compromised (S5B, S5D and S5E Fig). SynA mislocalization correlated with the spreading of the endocytic collar beyond the hyphal tips (Fig 7E, middle panel) and with the frequent loss of apical dominance (Fig 7E, bottom panel). Upon prolonged incubation hyphae arrested growth, the endocytic collar was completely disorganized, hyphal tips swelled and MyoE is delocalized to the cytosol (S5F–S5H Fig).

Even though fimbrin localizes exclusively to endocytic patches, Lifeact-GFP was only occasionally detected at the position of the SPK in *fimA::Tn341* tips. Slow growth of the mutant resulting from the endocytic/recycling defect could plausibly result in less secretory vesicles (SVs) arriving at the apex. This possibility was supported by the observation that SVs labelled with the R-SNARE SynA normally accumulating at the SPK in the wild type were not visible in the mutant, indicating that the SPK was indeed affected (S5A and S5B Fig). However, as a weaker accumulation of SVs might be obscured by the plasma membrane pool of SynA, we used myosin-5 MyoE, which is a component of the SPK, to identify the region at which Lifeact-GFP should be expected, and adjusted exposure times to maximize detection of the latter. Notably, in most *fimA::Tn341* tips MyoE localized to a crescent at the apical dome, rather than to an apical spot. Amongst the thirty hyphal tips examined 26 tips displayed MyoE as a crescent, but only in half of these were we able to detect an apical signal of Lifeact-GFP, and this was markedly less bright than in the wild-type (Fig 7E and 7F). Therefore *fimA::Tn341* impairs the apical accumulation of F-actin diagnostic of the SPK.

At the level of actin patches, impairing *fimA* function had two notable consequences. Firstly, actin was recruited to patches, but the lifetime of these augmented to 27.6 sec ± 5.4 S.D., *i.e.,* approximately twice that determined for the wild-type (Fig 7G and 7H). Most notably, patches were static, completely lacking any motile phase, as reflected by kymographs (Fig 7G and 7H) and time-lapse movies (S17 Movie). We conclude that proper cross-linking by fimbrin is required by dendritic actin in endocytic patches to power a productive endocytic event.

## Cofilin is essential for viability

Cofilin is the main F-actin severing protein in eukaryotic cells [61,62]. Deletion of *cof1*, the only gene encoding cofilin in *A. nidulans* (AN2317) is lethal (Fig 8A). Haploid *cof1Δ* conidiospores, obtained from heterokaryons, establish polarity before arresting growth with a typical terminal phenotype of endocytosis-less mutants (Fig 8B), with F-actin accumulating in cortical patches that do not progress any further (Fig 8C). Cof1-GFP appears to be lethal, but we could rescue the corresponding gene in diploids carrying a backup copy of the wild-type allele. In these heterozygous diploids, cofilin localizes to patches, enriched in the endocytic collar (S6A Fig) and, weakly, to the SPK, as shown by colocalization with FM4–64-labelled SVs (Fig 8D; note the strong labeling of mitochondria with FM4–64) The lifetime of cofilin in these patches was 20 ± 4.4 s S.D. (S6B Fig).

## Endocytic role of suppressor of RAS valine (Srv2)/cyclase associated protein (CAP)

As cofilin is essential and we wanted to test the effects of impaired actin turnover, we targeted one of the components of the complementary mechanism for disassembling actin filaments, which is mediated by Srv2/CAP and twinfilin. Srv2/CAP dramatically accelerates depolymerization of actin filaments at their pointed ends [63,64]. In *Aspergillus*, ablation of Srv2/CAP (AN0999) results in a severe phenotype markedly impairing colony growth (Fig 9A), as expected for a gene playing

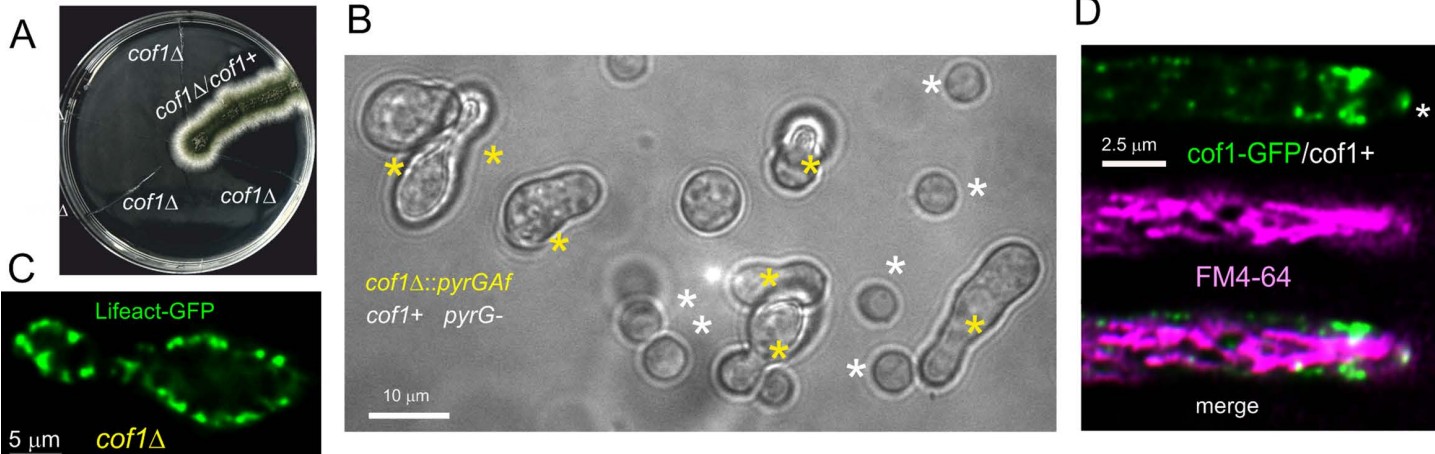

**Fig 8. Cofilin is essential. (A)** Heterokaryon rescue test. Haploid conidiospores obtained from heterokaryotic mycelia carrying *cof1 +pyrG89* and *pyrG89 cof1Δ::pyrG Af* nuclei were streaked on medium without pyrimidines, which selected against the wild type. Haploid *pyrG89 cof1Δ::pyrG Af* streaks did not give rise to visible growth, whereas a heterozygous diploid used as a control grew normally. Note that spores acquire a single nucleus during conidiogenesis, thereby resolving heterokaryosis. **(B)** Terminal phenotype with which *cof1Δ* germlings arrest growth, after 18 hours of incubation at 25°C. Yellow asterisks indicate *pyrG89 cof1Δ::pyrG Af* germlings with the characteristic yeast-like phenotype, whereas white asterisks indicate *pyrG89 cof1+* conidia unable to establish polarity. **(C)** Localization of Lifeact-GFP in *cof1Δ* mutant germling. The image is a middle plane of a deconvolved Z-stack. **(D)** Localization of endogenously tagged Cof1-GFP in heterozygosis with a wild-type allele in a diploid strain. Note the presence of cofilin in the actin collar. In addition, there is detectable Cof1-GFP fluorescence at the position of the SPK (asterisk), which is labeled with FM4-64. Note that in *A. nidulans* this structure is only weakly and occasionally labeled with this dye, which concentrates strongly in the mitochondria. The image is made with middle planes of deconvolved z-stacks.

an important role in endocytosis. At the cellular level, Lifeact-GFP imaging revealed that *srv2Δ* disorganizes the normal distribution of actin patches such that either they no longer assemble into a collar or they gather in an abnormal collar that extends relatively far away from the apex (S7A Fig). Mutant hyphae frequently undergo branching in the tip cell, indicating loss of apical dominance (S7A Fig). In view of these phenotypes we examined the integrity of the SPK in the *srv2Δ* background using the SPK resident MyoE as reporter. The mutation resulted in the almost complete delocalization of MyoE to the cytosol. A minor proportion of MyoE localized to the tips, often forming an apical crescent (Fig 9B), which in some examples could be resolved as individual dots (S7B Fig). Lifeact-GFP signal colocalizing with the MyoE crescent was undetectable in 40 out of 43 tips examined, indicating that the absence of this protein disorganizes directly or indirectly the SPK. Kymograph analysis of movies acquired at 5 fps revealed that *srv2Δ* results in patch lifecycles that double in duration the wild-type, although these patches still transited through static and motile phases (Fig 9C).

A useful estimation of the degree to which endocytosis is prevented by any given genetic intervention is determining to what extent is the efficiency of the endocytic collar impaired. This may be achieved by measuring the polarization of protein cargo that becomes polarized by endocytic recycling [7,65]. In the wild type, endogenously tagged chitin synthase B (ChsB, an integral membrane protein) localizes to the SPK and to the apical dome plasma membrane, reaching, on average, a distance of approximately 2 µm from the apex before being internalized by the endocytic collar. In this ChsB endocytic recycling assay, *srv2Δ* severely impaired this efficiency, which resulted in diffusion of ChsB far away from the hyphal tips (Fig 9D). Thus, Svr2 is required to determine the normal duration of the actin burst during endocytic internalization and to maintain the coupling of the endocytic collar to the growing tip.

As endogenous tagging of the protein is debilitating, indicating incomplete function (Fig 9A), the behavior of Srv2-GFP was uninformative, beyond the observation that it is able to localize to cortical patches (S7C Fig). These Srv2-GFP patches had a lifetime longer than one minute, and underwent short-distance movements similar to those observed with Cof1-GFP (S7D Fig shows kymographs comparing Srv2-GFP to Lifeact patches).

Once the involvement of Srv2 in endocytosis had been determined, we proceeded to analyze the effects of its ablation on the dynamics of actin. Time-lapse Movies of Lifeact in *srv2Δ* cells revealed the appearance of bright actin structures that often connected patches localizing to opposite sides of the hyphae (S18 Movie), supporting the notion that F-actin disassembly is indeed deficient in *srv2Δ* cells. These Movies also revealed the presence of cortical structures that moved basipetally and gave rise to "actin worms" akin to those observed in the wild type, but which appeared to be more abundant in the *srv2Δ* mutant (S19 Movie and S7E Fig). Movies additionally confirmed that "actin worms" originated at actin patches (S20 Movie) and that they traveled for several micron-long distances before being depolymerized. In the example shown on Fig 9E one such worm was filmed leaving behind a tail of F-actin that resembles that left by *Listeria* cells (readers should also consult S21 Movie). Depolymerization of these "actin worms" took place from the tip proximal pole, suggesting that this pole represents the barbed end of filaments originating at endocytic patches (depolymerization by the pointed end was predictably impaired by the absence of Srv2). The interpretation that the tip-proximal (lagging) end is the barbed end implies that this movement cannot be attributed to treadmilling, which would require the opposite polarity.

## Arp2/3 mediates actin polymerization in endocytic patches but is absent from exocytic actin structures

The burst of F-actin polymerization driving endocytosis reflects the formation of branched actin networks synthesized by the Arp2/3 complex [14,66]. Hyphal tip growth might also require bursts of branched actin synthesis at the apex, also driven by Arp2/3. To study this possibility, we tagged endogenously the ARPC5 (AN4919) and ARPC1 (AN5778) subunits with GFP (for Arp2/3 we will use *S. pombe* nomenclature). *arpC5-gfp* resulted in lethality, as did *arpC1Δ* (Fig 10A). *arpC1Δ* spores collected from heterokaryons arrested growth with the characteristic morphology of abortive germtubes unable to maintain polarity (Fig 10A, boxed). In contrast, endogenous GFP tagging of ArpC1 does not affect colony growth, indicating that the fusion protein is functional (Fig 10B). ArpC1-GFP is present in patches akin to those labeled by chimeric actin reporters (Fig 10C) and (S22 Movie). The lifetime of ArpC1 in these patches was 13.9 sec +/– 1.9 S. D. (Fig 10D), which

                                    

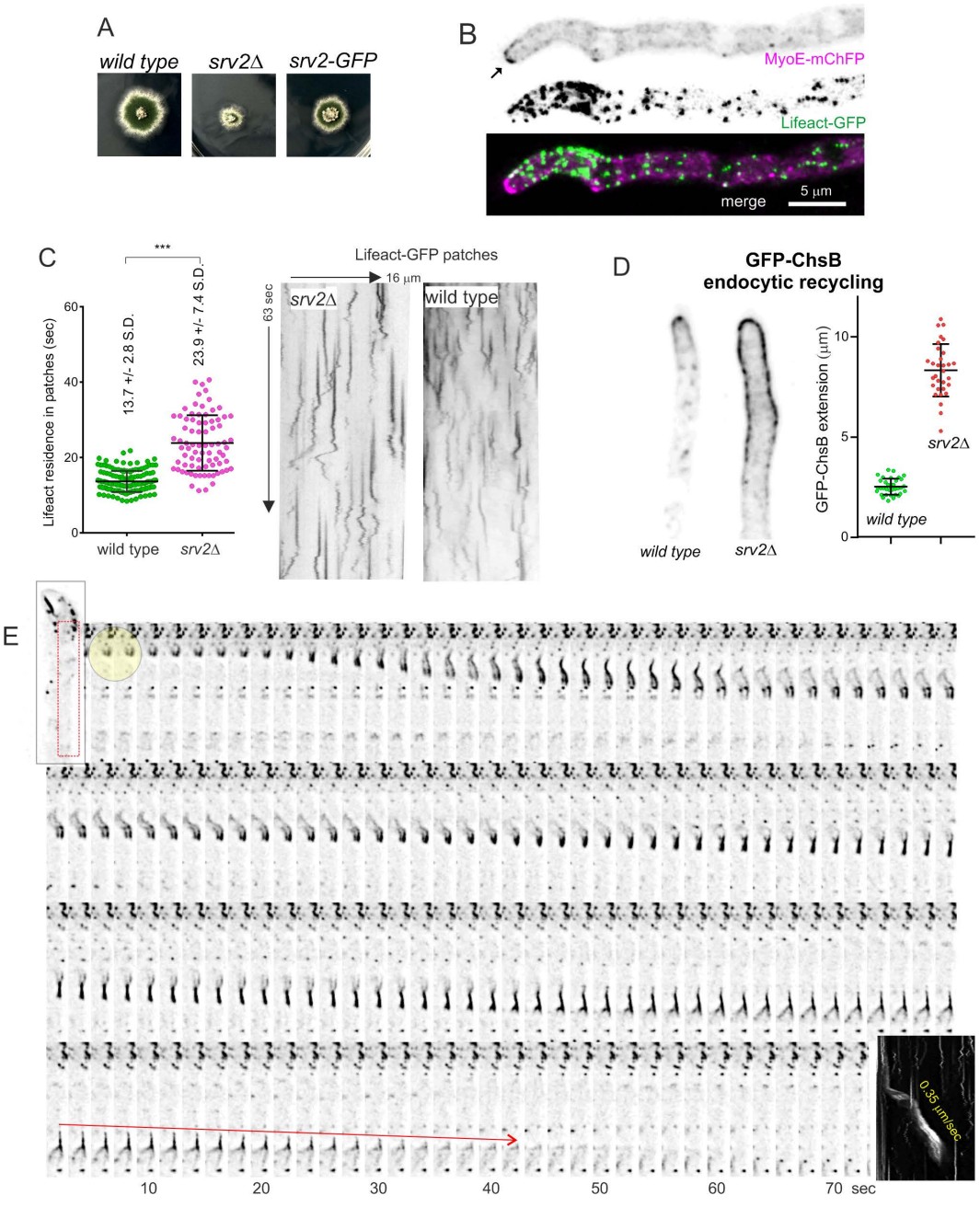

**Fig 9. The role of Srv2/CAP in endocytosis. (A)** Growth tests on minimal medium at 37°C showing that *srv2Δ* markedly impairs colony growth, with the endogenously GFP-tagged allele resulting in an intermediate growth phenotype, indicative of partial-loss-of-function **(B)** Lifeact-GFP imaging shows that *srv2Δ* disrupts the normal distribution of actin patches, delocalizing actin from the SPK, which correlates with relocalization of the SPK resident MyoE to the cytosol and to a weakly labeled apical crescent. Images are MIPs of deconvolved Z-stacks. **(C)** Kymograph analysis of wild-type and *srv2Δ* time-lapse sequences of Lifeact-GFP acquired at 5 fps. The two data sets were significantly different according in a Mann-Whitney test. **(D)** Endocytic recycling of GFP-ChsB: in the wild-type, this cargo of the endocytic recycling pathway is polarized by endocytosis to the approximately 2 μm region from the apex, mostly corresponding to the apical dome. In contrast, in *srv2Δ* hyphae, GFP-ChsB is spread several microns away from the apex due to the endocytic deficit of the mutant. Error bars, mean ± S.D. **(E)** This F-actin worm appears to consist of 2 parallel filaments seemingly propelled by forces originating at the tail. The box on the right shows the corresponding longitudinal kymograph indicating that these filaments move at an average speed of 0.35 μm/s. The red arrow marks the successive positions of the trailing end of this actin worm.

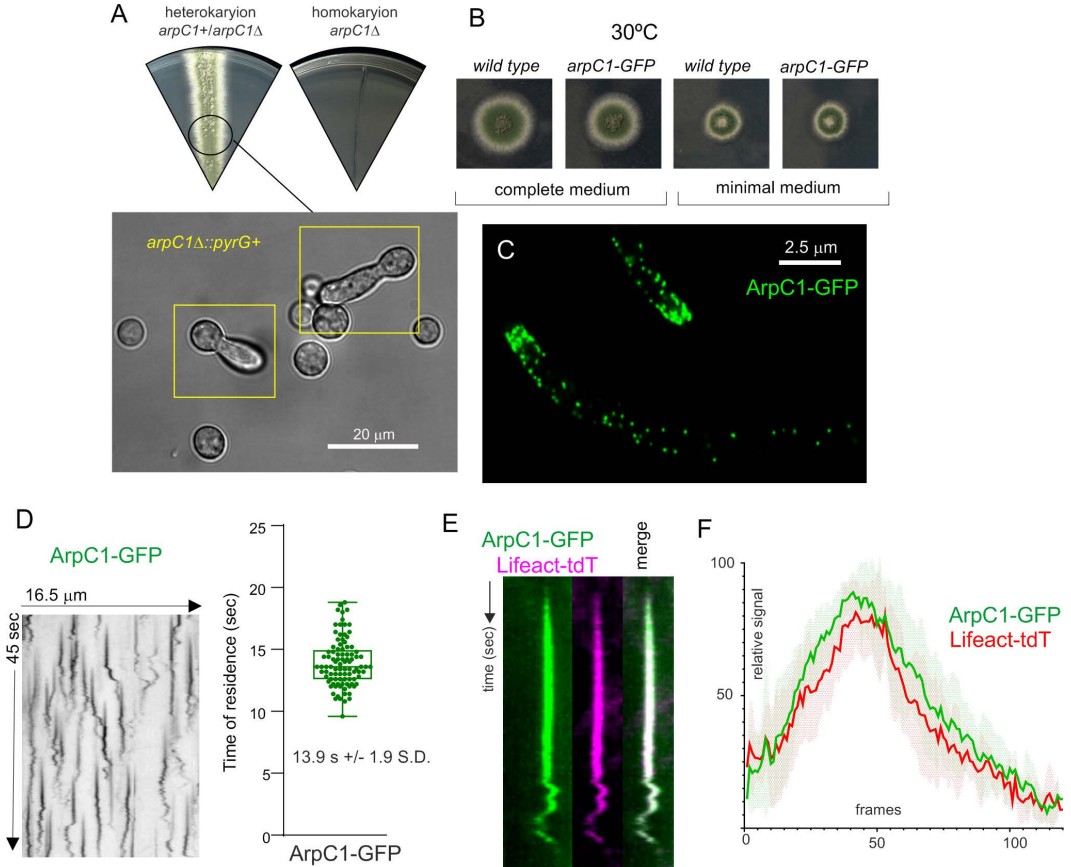

**Fig 10. The Arp2/3 complex arrives at actin patches at roughly the same time than F-actin.** (A) *arpC1Δ* ablating a key subunit of the Arp2/3 complex results in lethality. The Nomarski image shows that *arpC1Δ* spores, collected from heterokaryotic mycelia, give rise to germlings that arrested growth shortly after establishing polarity, with the typical morphology of cells deficient in endocytosis. (B) Growth tests showing that endogenously tagged ArpC1 (*arpC1-GFP*) is functional (C) ArpC1-GFP localizes to actin patches. MIP of a deconvolved Z-stack. (D) Kymograph analysis: ArpC1-GFP resides in actin patches for approximately 14 sec. Error bars are mean±S.D. (E) ArpC1-GFP and F-actin (Lifeact-tdT) colocalize across time. (F) Analysis of the relative signals of ArpC1-GFP and Lifeact-tdT in *N*=5 patches. Each frame corresponds to 200 msec of a time-lapse series with simultaneous acquisition of the two channels using a Gemini beam splitter.

matches those of AbpA[Abp1] and Lifeact. Indeed, the lifecycles of ArpC1-GFP and Lifeact-tdT were essentially indistinguishable (Fig 10E and 10F), in agreement with the fact that the burst in actin polymerization driving endocytosis is catalyzed by Arp2/3.

A key observation was the complete absence of Arp2/3 at the position of the SPK. This finding is important because it implies that the mesh of actin present in this structure is not branched actin. To buttress this key conclusion further, we used fluorescent versions of the SPK residents MyoE (myosin-5) and SepA (formin). Fig 11A shows that MyoE is present in a spot at the apex [51], whereas vesicles that arrive at the tip by the endocytic recycling pathway, labeled with the endocytic tracer FM4–64 [23,49], show a slightly wider distribution, overlapping that of the secretory vesicle marker RabE[RAB11] [36,67]. Myosin-5 is a barbed end-directed motor that in the steady-state accumulates at the SPK because SepA, the only *A. nidulans* formin, nucleates actin filaments with their barbed ends at the SPK [5,39,51,68]. To get a robust signal of SepA, we labeled this protein endogenously with a triplicated GFP. Myosin-5 colocalizes strictly with the formin in the apices of hyphae (Fig 11B), validating it as SPK marker. Fig 11C is a confocal image showing that Arp2/3 is completely

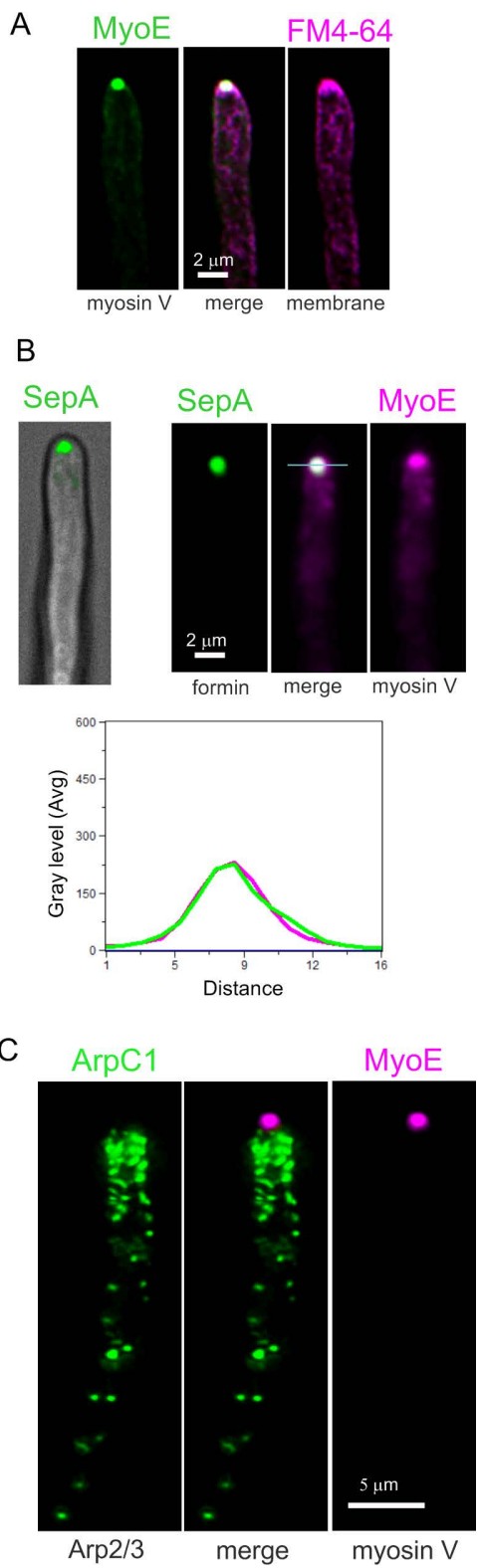

**Fig 11. The Arp2/3 complex is excluded from the SPK. (A)** MIPs showing the localization of MyoE-GFP to the apex, whereas secretory vesicles labeled with FM4-64 display a slightly broader distribution at the tip. **(B)** Left, composite confocal image showing the localization of the SepA formin to

the apex. Right, SepA and MyoE colocalize at the SPK. **(C)** Arp2/3, labeled with ArpC1-GFP, is absent from the SPK, labeled with MyoE-mChFP. Images are MIPs of confocal Z-stacks.

absent from the hyphal dome, restricting its localization to the endocytic collar, whereas MyoE is completely absent from the endocytic collar, restricting its localization to the SPK. Thus, the SPK does not contain branched actin.

## Verprolin plays an important role early during the F-actin lifecycle, being required for the transition between motile and non-motile phases

Verprolin coordinates the nucleation promoting factor (NPF) activities of WASP/Wsp1 and type I myosin during Arp2/3-mediated branched actin polymerization [69–72]. While its interacting network has been studied, its physiological role is insufficiently understood. There is a single verprolin homologue in *A. nidulans*, encoded by *vrp1* (AN1120). Ablation of Vrp1 results in a marked inhibition of growth, consistent with it playing an important role in endocytosis (Fig 12A). We verified that endogenous C-terminal tagging of Vrp1 does not impair function before analyzing its localization and dynamics. Vrp1-GFP localizes to actin patches/endocytic collar but is absent from the SPK (Fig 12B), further indicating that this structure does not contain branched actin. The average time of residence of Vrp1 in these patches is $8.1 \pm 0.8$ sec (S.D), which is markedly shorter than that of Arp2/3 (Fig 12C). Dual color imaging established that Vrp1 is recruited to patches on the first half of the Arp2/3 lifecycle, during the non-motile phase (Fig 12D). S23 Movie shows that Vrp1-GFP is recruited to actin patches with ArpC1-mChFP and disappears from them before actin patches transit from the non-motile to the motile phase.

Kymograph analysis of Lifeact patches revealed two interesting observations: one is that the absence of Vrp1 completely abrogated the motility of actin patches across the whole F-actin cycle, indicating that it is required for the transition from the non-motile to the motile phase (Fig 12E). The second, that the absence of Vrp1 extended the lifecycle of F-actin to $25.7 \pm 4.1$ sec (S.D.) (Fig 12F), approximately doubling the lifetime in the wild-type. So, Vrp1 plays an important role in generating the branched actin network that powers endocytic internalization. In the endocytic recycling assay, ChsB was markedly depolarized in the *vrp1*Δ background, diffusing across the plasma membrane up to 15 μm away from the apex, in sharp contrast to the wild-type, in which ChsB extended for *circa* 2 μm (Fig 12G). Therefore, this functional analysis combined with growth tests and studies of F-actin residence in endocytic patches underline the key importance that verprolin plays in endocytosis by regulating F-actin dynamics.

## *abpA*Δ and *capA*Δ show a strong synthetic negative interaction in growth and endocytosis

Several physiological systems curb F-actin polymerization. *capA* (AN2126) [73] encodes the alpha subunit of capping protein, which in Arp2/3 networks restricts the addition of actin subunits onto the barbed ends of actin filaments [12,74], arresting polymerization. Ablation of capping protein results in only mild endocytic defects, and it has been reported that in the budding yeast this results from the existence of an alternative pathway mediated by Aim3 and Abp1, with the simultaneous absence of both systems making the endocytic defect severe [74]. We set out to study this mechanistic redundancy in *Aspergillus*, taking advantage of the key role of endocytic recycling in hyphal growth. AbpA localizes to the endocytic collar but not to the SPK, indicating that it plays no exocytic roles (Figs 6A and 13A). *abpA*Δ strains have no growth defects [5,6], whereas *capA*Δ strains display a marked impairment of growth (Fig 13B), attributable, in part, to its endocytic role, but also to its anticipated role in the regulation of non-endocytic F-actin structures, such as filaments/cables emanating from the SPK. Compared to *abpA*Δ and *capA*Δ single mutants, the double mutant is severely debilitated (Fig 13B and 13C), indicating that CapA and AbpA[Abp1] share a nearly essential function.

The debilitating effect caused by *capA*Δ is strong, but does not compromise viability. Endogenous GFP-tagging of CapA results in a slight growth defect at 25–30°C (endocytic assays were carried out within these temperature limits), which

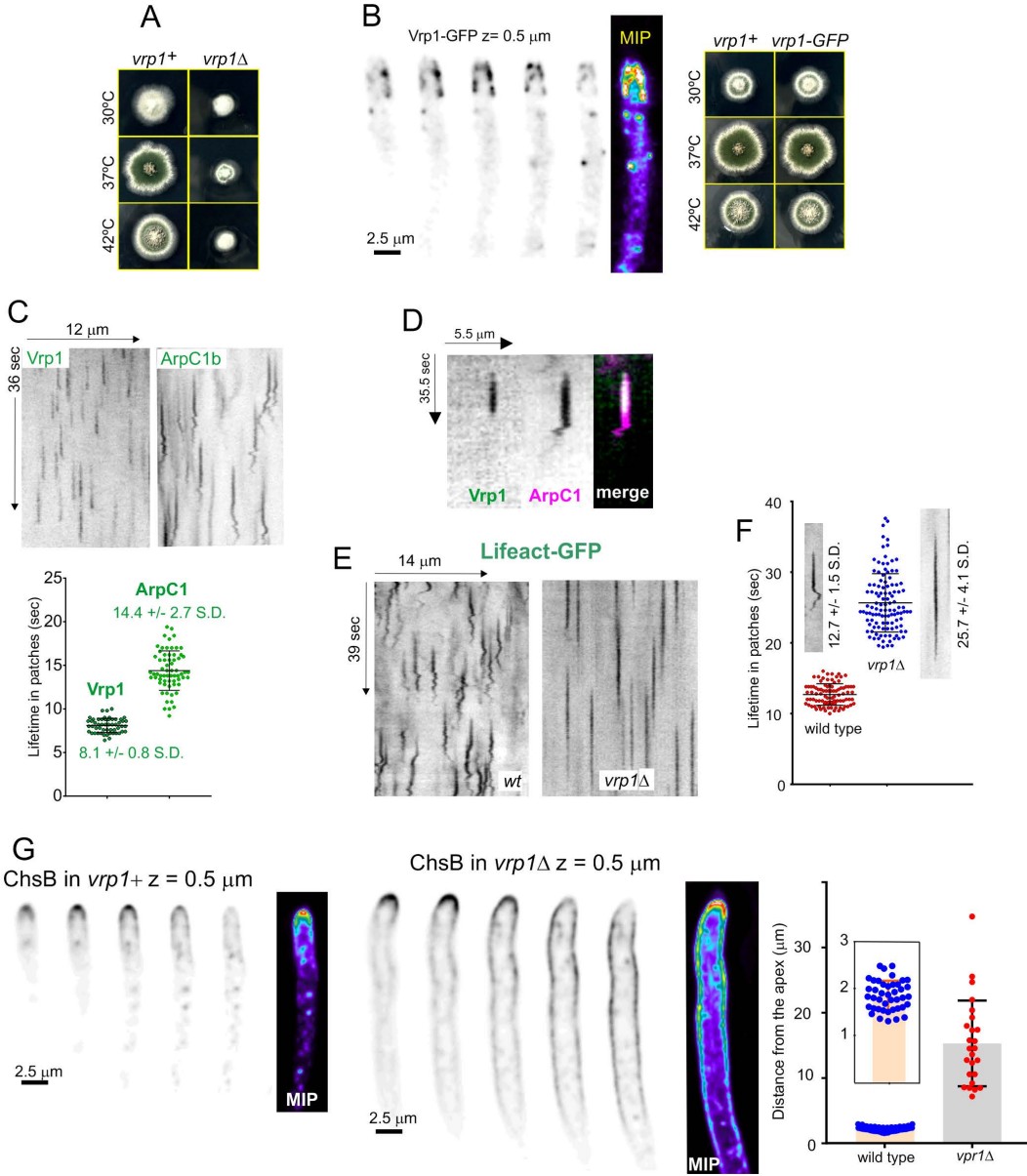

**Fig 12. The motile phase of actin patches is dependent on Vrp1. (A)** Marked growth defects of *vrp1Δ* cells cultured on minimal medium. **(B)** Endogenously tagged Vrp1-GFP is functional and localizes to actin patches. Consecutive sections of a z-stack and MIP (colored, heat LUT). **(C)** Kymographs comparing the behavior of Vrp1-GFP with ArpC1-GFP, derived from time lapse acquisitions at 5 fps. The lifetime of Vrp1-GFP is approximately half that of ArpC1-GFP. **(D)** The presence of Vrp1 on actin patches is restricted to the first half of the actin module. **(E)** and **(F)** Kymographs showing that the lifetime of Lifeact-GFP in *vrp1Δ* cells doubles that of the wild-type, and that *vrp1Δ* patches lack the motile phase. **(G)** Z-series and MIPs of GFP-ChsB in *vrp1Δ* compared to the wild-type show the markedly longer distance to which ChsB diffuses across the membrane, indicative of a major defect in endocytic recycling. Error bars are mean ± S.D.

becomes conspicuous at 37° (S8A Fig). Despite this caveat, CapA-GFP localized to endocytic patches, but the time of residence of CapA-GFP is 27.3 sec ± 8.7 S.D.) (S8B Fig). Because capping protein is expected to overlap largely with the lifecycle of F-actin, this long time of residence, which doubles that of F-actin, almost certainly reflects impaired function of the fusion protein.

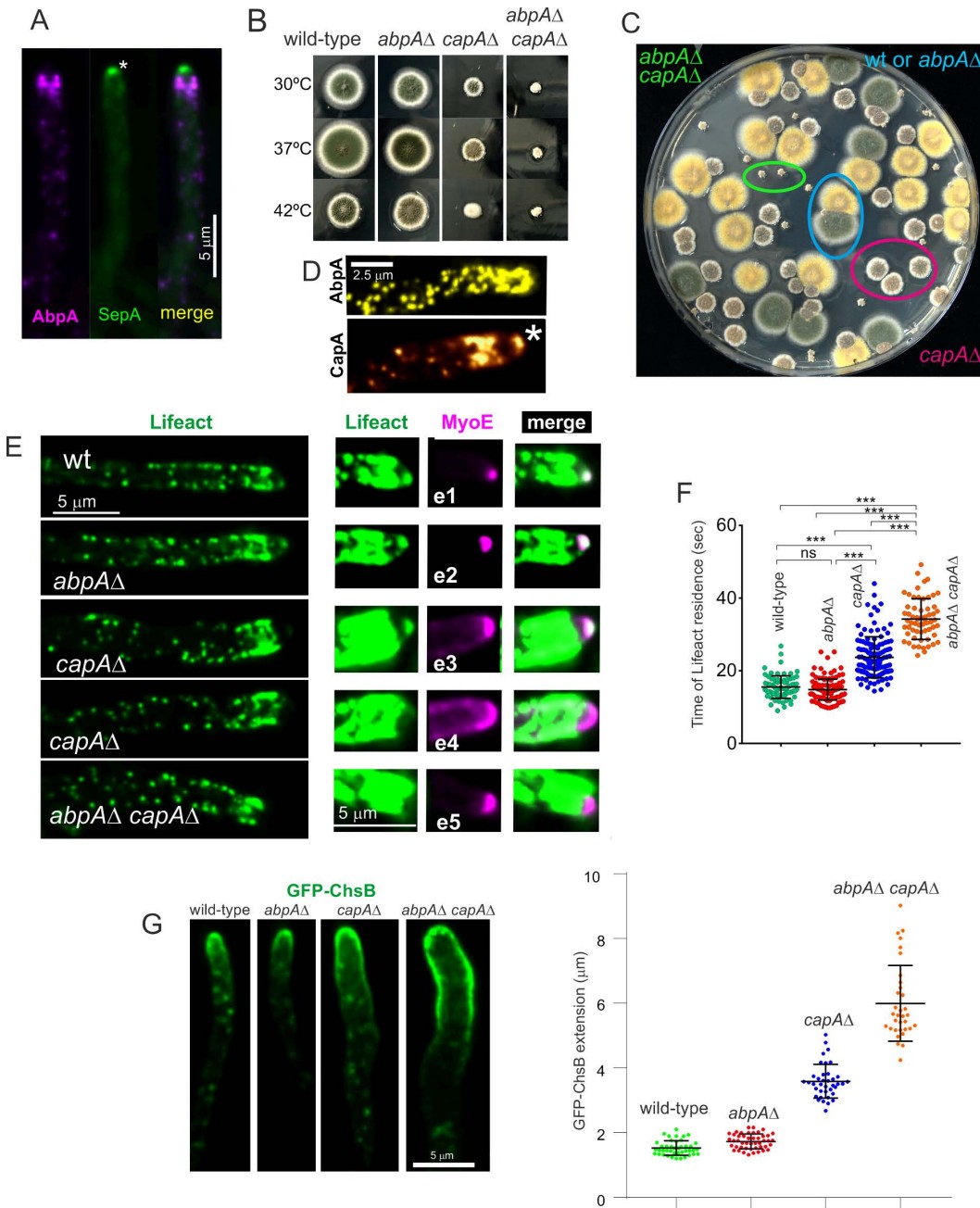

**Fig 13. Synthetic negative interaction of *abpA* Δ and *capA*Δ mutants. (A)** AbpA[Abp1]-mRFP localizes to the endocytic collar but not to the SPK, labelled with SepA. The image is a MIP. The asterisk indicates the SPK. **(B)** Growth tests showing that *abpA*Δ, which by itself does not cause any notice-able defect, shows a strong synthetic negative growth interaction with *capA*Δ. **(C)** Progeny of a heterozygous cross between *abpA*Δ (indistinguishable from the wild-type, cyan circle) and *capA*Δ (circled in red) parental strains was spread on a plate, highlighting the severe growth impairment in the *abpA*Δ *capA*Δ double mutant (circled in green). **(D)** CapA-GFP (golden LUT) localizes to the SPK (indicated with an asterisk), whereas AbpA[Abp1]-GFP (yellow LUT) does not. **(E)** Maximum intensity projections (MIP) of Lifeact-GFP in strains that co-express MyoE-mChFP to localize the SPK. Left images display Lifeact-GFP with normal contrast. The right panels focusing on the tips are overcontrasted in the green channel to maximize the possibility of detecting the SPK. Composites e1 and e2 depict that *abpA*Δ shows a normal SPK onto which MyoE and Lifeact colocalize. *capA*Δ results in a heterogeneous phenotype: in all cases MyoE formed a crescent rather than localizing to an apical spot, but only in 50% of these mutant tips a signal of Lifeact-GFP was weakly detectable. *abpA*Δ *capA*Δ double mutants showed MyoE as an apical crescent, without detectable Lifeact signal. **(F)** Quantitation of the time of residence of F-actin in endocytic patches of the indicated strains. Data were analyzed using the Kruskal-Wallis test with Dunn's multiple comparisons. Error bars are mean±S.D. **(G)** Diffusion across the plasma membrane of GFP-ChsB is normal in the single *abpA*Δ mutant, significantly increased in the single *capA*Δ mutant and markedly elevated in the double mutant, indicating progressively increasing endocytic deficit. Plots depict the average±S.D.

Besides actin patches, CapA-GFP localizes to the SPK (Fig 13D), suggesting that it plays a role in this structure whose deficiency might contribute to the phenotype. Indeed, single *capAΔ* and double *capAΔ abpAΔ* (but not single *abpAΔ*) mutations apparently delocalize F-actin from the SPK (Fig 13E, left row). To investigate the possibility that F-actin had been delocalized from the SPK without actually disrupting the structure we used MyoE-mChFP. Co-filming of Lifeact and MyoE confirmed that *abpAΔ* does not affect the SPK, with the two reporters colocalizing like in the wild-type (Fig 13E, e1 and e2). In contrast, *capAΔ* clearly disrupted this structure. Rather than concentrating on a sharp spot, MyoE formed a crescent of variable width. In those tips with a narrower crescent, increasing the contrast of the images enabled detection of a weak signal of Lifeact located in the middle of the crescent (Fig 13E, e3). In roughly half of the population, however, the crescent of MyoE extended across the whole apical dome without any actin signal that might suggest the presence of a residual SPK (Fig 13E, e4). In *capAΔ abpAΔ* double mutants, the whole population showed MyoE as a crescent without any overlapping Lifeact signal (Fig 13E, e5).

We conclude that the absence of *capA* disrupts the SPK. We acknowledge, however, that this disruption might simply result from the decreased arrival of SVs to the apex due to the reduction in endocytic recycling (see below), which is supported by the observation that all *abpAΔ capAΔ* tips show an extended MyoE crescent devoid of actin (Fig 13E, e5), even though AbpA is absent from the SPK. That the disruption of the SPK results, at least in part, from a reduction in the number of SVs arriving at the apex is further supported by experiments using GFP-ChsB, which localizes to SVs that recycle cargo from the TGN. These SVs accumulate in the SPK, which is conspicuously visible in the wild-type and in the *abpAΔ* mutant. In contrast, it is barely visible in the *capAΔ* single mutant and is completely absent in the double *capAΔ abpAΔ* (S8C Fig).

*capAΔ* impaired the F-actin cycle in patches, nearly doubling the time of residence in the wild-type (from 14.8 sec to 23.7±5.6 (Fig 13F). Combining *capAΔ* with *abpAΔ* resulted in a further increase in Lifeact time of residence to 34.2±5.6 sec (Fig 13F). Thus, the time of residence of Lifeact in patches correlates inversely with growth, strongly indicating that an increase in this parameter reflects a larger defect in endocytosis. We validated this conclusion by determining the effects of the mutations on the endocytic recycling of GFP-ChsB. These experiments (Fig 13G) demonstrated that endocytosis is essentially normal in the *abpAΔ* single mutant, significantly less efficient in the *capAΔ* single mutant and severely debilitated in the *capAΔ abpAΔ* double mutant. In agreement with this marked defect in endocytic recycling, *capAΔ abpAΔ* hyphae displayed recurrent losses of polarity, abnormal hyphal morphogenesis, major disorganization of the actin cytoskeleton, ectopic septation and essentially immotile actin patches (S8D and S8E Fig and S24 Movie).

## Formin is not essential for the biogenesis of endocytic F-actin

Because the Arp2/3 complex is unable to nucleate new actin filaments without a mother filament that serves as seed for the formation of branches, there must be a mechanism/mechanisms ensuring that the priming filament is assembled at, or recruited to the sites of endocytosis. This aspect of the biogenesis of actin patches is insufficiently understood in general and has never been investigated in filamentous fungi. It has been claimed that this mother filament is provided by formin synthesizing such a filament *de novo* at endocytic sites. However, the single *A. nidulans* formin, SepA, localizes to the SPK and to contractile rings, but has not been detected in the position of the endocytic collar, despite the high density of actin patches that it contains [39,50,53]. As it could be argued that this lack of detection is due to the low abundance of this protein in the cell, we decided to address this problem genetically, exploiting *sepA1*, a heat sensitive mutation in *sepA*, which inactivates function at 37°C (Fig 14A) Indeed, whereas the localization of actin at the SPK was not affected by shifting wild type cells to 37°C, the signal of actin at this locale was completely lost when *sepA1* cells were filmed instead (Fig 14B). Under these conditions, the average lifetime of Lifeact in endocytic patches was essentially indistinguishable at 28°C or 37°C, irrespective of whether cells were wild type or carried the *sepA1* mutation, strongly arguing against involvement of SepA in the biogenesis of actin patches (Fig 14C). To buttress this conclusion, we repeated similar experiments using endogenously tagged FimA-mChFP, which led to the same conclusion (Fig 14D). We conclude that formin, at least

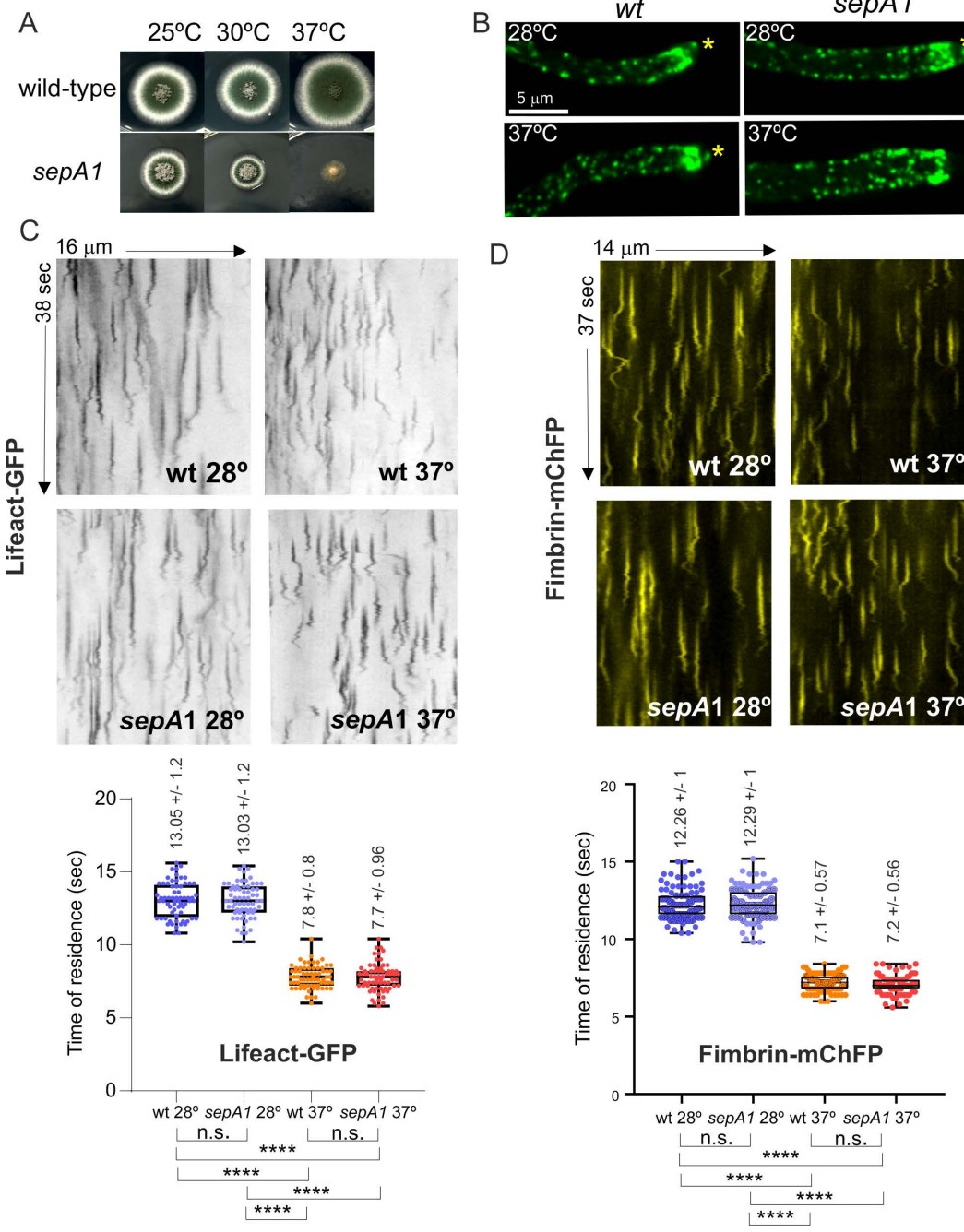

**Fig 14. Conditional inactivation of formin does not affect the lifecycle of actin in patches. (A)** Growth tests on complete medium showing the heat sensitivity growth phenotype of sepA1 cells. **(B)** Delocalization of F-actin from the SPK (indicated with an asterisk) in cells carrying the *sepA1 ts* mutation at the restrictive (37°C), but not at the permissive (30°C) temperature (see also Fig 16B). **(C)** Kymographs (5 fps) of Lifeact-GFP at the indicated temperatures and mutant conditions analysis. Graphs on the right show the corresponding quantitative analyses of time of residence. Box and whiskers error bars are minimal to maximum; datasets were analyzed by ANOVA followed by Tukey's multiple comparison test; n.s., non-significant; ****, P<0.001. **(D)** As above, but using FimA-mChFP as reporter.

in the presence of an intact endocytic machinery, is not essential to provide mother filaments that prime Arp2/3-mediated synthesis of branched actin networks during endocytosis.

## Dip1 funnels actin towards endocytic patches

In view of this lack of involvement of formin, we turned our attention to the *Aspergillus* homologue of Dip1, a member of the WDS (WISH/DIP/SPIN90) family of proteins. Unlike WASP and other NPFs that recruit Arp2/3 to the sides of pre-formed actin filaments, Dip1 binds and activates Arp2/3 without a pre-existing mother filament, remaining bound to the nascent linear filament by Arp2/3 [15–17,75,76]. Therefore, once a key role of formin had been discounted, Dip1 was the alternative candidate to 'prime' endocytic patches. Endogenously GFP-tagged Dip1 is functional (S9A Fig) but weakly fluorescent. Dip1-GFP localizes to actin patches (Fig 15A), in agreement with its predicted role in endocytosis. Ablation of Dip1 results in a marked growth defect, but *dip1Δ* cells are viable, indicating partial functional redundancy of Dip1 with other potential providers of actin patch seed filaments (Fig 15B). Kymographs revealed that Lifeact residence time in patches nearly doubled the wild-type (23 vs. 14 sec, Fig 15C for *dip1Δ* and 3E for the wild-type), which was confirmed using FimA-GFP (26 vs. 14.7 sec in the mutant vs. the wild-type) (S9B Fig). The key role that Dip1 appears to play in endocytosis was established using endocytic recycling assays of chitin synthase B, which showed that the distance to which this transmembrane cargo diffuses away from the tip in *dip1Δ* cells is the highest among any other mutant tested (Fig 15D). Endocytic recycling and hyphal tip growth are intimately related, and therefore it should come as no surprise that the apical extension rate of *dip1Δ* is reduced by half relative to the wild-type (S9C Fig). We conclude that Dip1 is important yet nonessential for endocytosis, demonstrating that there must be a mechanism to provide mother filaments for Arp2/3 that is independent of Dip1.

Even though *dip1Δ* does not affect the localization of the SepA formin, the MyoE myosin or Lifeact-GFP in the SPK (Fig 15E–15G), the most remarkable effect of the mutation is that it dramatically facilitates visualization of Lifeact-labeled actin cables, which become very noticeable in the tips (Figs 15F–15I and S9D). Inspection of images immediately provided a plausible explanation: actin patches appeared to have been removed from the tip proximal regions and the endocytic collar was completely absent, which on the one hand facilitates detection of cables without the strong background of actin patches and on the other very likely enlarges the pool of G-actin available for linear F-actin structures [77]. Time lapse sequences in cells ablated for Dip1 revealed a remarkable flow of actin cables extending from the SPK towards basal regions of the hyphae (Fig 15H and 15I). This flow was visualized by kymographs as tilted lines originating at the apex, and these kymographs together with S25 and S26 Movies demonstrate that these represent actin cables that are continuously emerging from the SPK, in all likelihood synthesized by the SepA formin. These cables are also detectable in *dip1Δ* tips using TpmA-GFP (S27 Movie and Fig 15J). Importantly, together with data described above, experiments with *dip1Δ* cells strongly indicated that cables originating at the SPK are different from the "actin worms": the former are restricted to the apex-proximal region, do not travel away from the tip and contain tropomyosin, whereas the latter, i.e., "actin worms" are present all across the hyphae, travel for long distances, are not detectable with TpmA-GFP (S10 Fig and S28 Movie) and appear to associate closely with endocytic patches Indeed, the lower density of endocytic events in *dip1Δ* cells facilitated visualization in apex-distant regions of numerous actin patches that appeared to be captured by actin worms, or that were being formed in close vicinity to these cables (S9E Fig).

## Formin does not appear to seed actin patches in the absence of Dip1

Analysis of *dip1Δ* kymographs showed that the front limit of actin patches roughly coincided with the basal end of cables emanating from the SPK (Fig 15F), which led us to hypothesize that cables synthesized by the SepA formin might be able to provide mother filaments for Arp2/3 when Dip1 is absent. This hypothetical model is inspired by the "touch and trigger" mechanism originally proposed by Pollard and co-authors for *S. pombe* [78]. In view of this possibility we reconsidered a potential role of SepA that would be masked by the presence of Dip1, but that would become important when the latter is

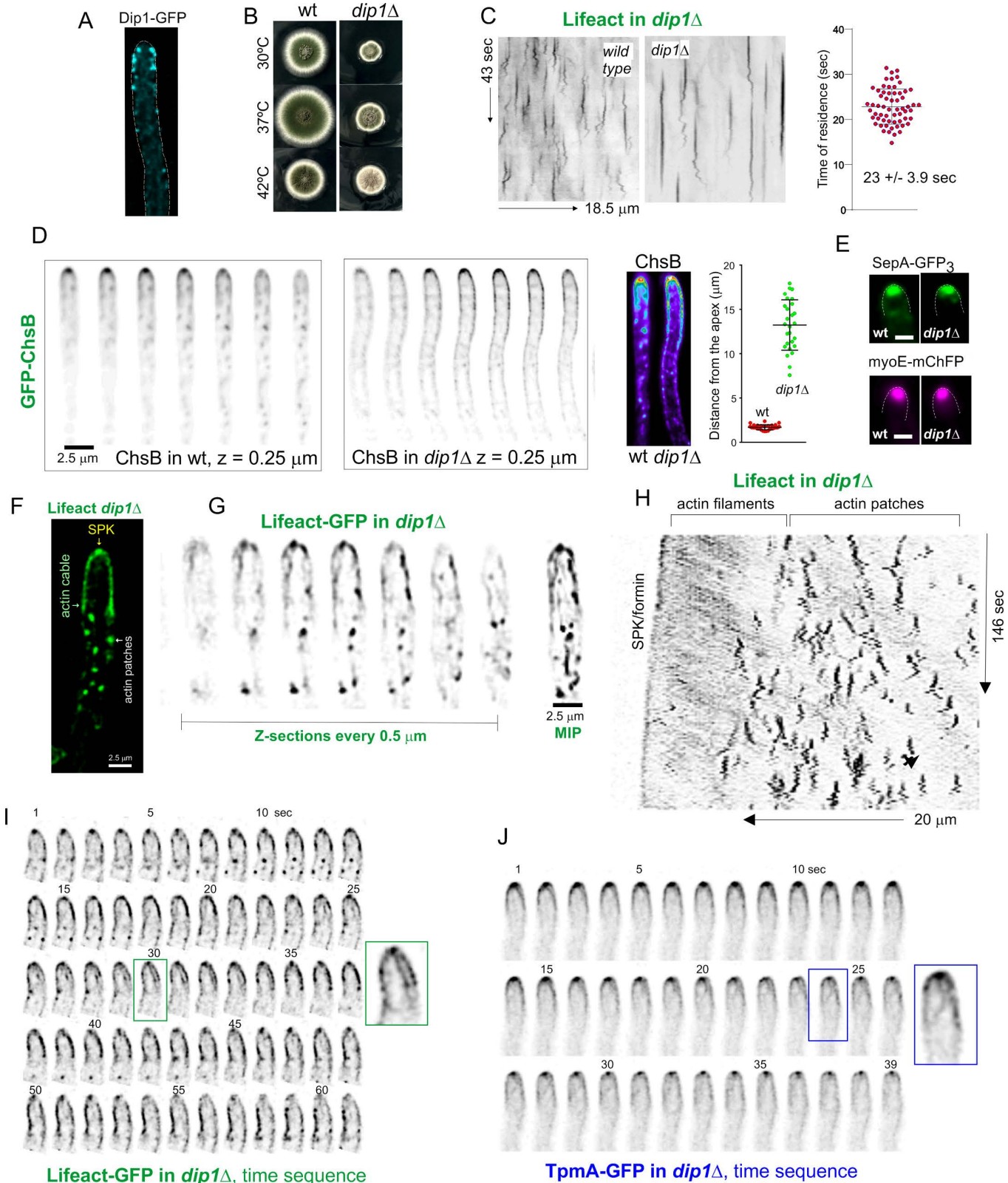

**Fig 15. Dip1 is a key regulator of endocytic F-actin. (A)** Localization of Dip1-GFP to the endocytic collar. MIP of a deconvolved Z-stack **(B)** *dip1Δ* results in a marked growth defect. **(C)** Analysis of Lifeact-GFP actin patches in *dip1Δ* cells (time resolution is 5 fps). Error bars are mean±S.D. **(D)**

Endocytic recycling of ChsB is markedly affected by *dip1Δ*, such that the reporter diffuses away from the tip for notable distances. The color images are maximal intensity projections represented with a heat LUT. Error bars in the graph are mean±S.D. **(E)** *dip1Δ* does not delocalize the formin SepA or the type V myosin MyoE from the SPK. Images are middle planes of deconvolved Z-stacks. **(F)** The actin collar is disorganized and actin patches appeared to be excluded from the region occupied by actin cables including the apical dome and subtending regions. **(G)** Prominent actin cables originating at the SPK become visible in the *dip1Δ* mutant. **(H)** Kymographs of Lifeact-GFP capture how actin cables originate regularly at the region of the SPK, appearing as diagonal lines whose basal ends roughly coincide with the region occupied by actin patches. **(I)** Frames of a time-lapse sequence of Lifeact-GFP showing the dynamics of actin cables emerging from the SPK in a *dip1Δ* cell (S25 Movie). **(J)** Frames of a time-lapse sequence of tropomyosin -GFP showing the dynamics of actin cables emerging from the SPK in a *dip1Δ* cell (S27 Movie).

absent. Notably, despite the fact that at 30°C both *dip1Δ* and *sepA1* behave as hypomorphs, resulting in reduced colony diameter in growth tests, their combination did not result in any synthetic growth defect (Fig 16A). At 37°C *sepA1* was severely debilitating, yet its combination with *dip1Δ* did not enhance this weakness any further, either (Fig 16A). While we acknowledge that *sepA1* is not a null allele, and that detection of synthetic negative interactions might be complicated by allele-specific phenotypes involving the multiple roles of formin, these experiments strongly suggest that if formin-mediated F-actin polymerization rescues the defect associated with the absence of Dip1 in endocytosis, it cannot be the only mechanism.

To gain further insight into the potential role of formin, we examined the effects of *sepA1* at the cellular level. Fig 16B demonstrates that *sepA1* completely disorganizes the SPK, visualized with MyoE-GFP$_3$, after a 15 min incubation at 37°C. We next analyzed *sepA1* cells expressing Lifeact-GFP that had been shifted from 20°C to 37°C. Despite that the endocytic collar is completely disorganized, actin patches were present all across the cells, including the tips, which appear swollen due to the expression of the mutation. Remarkably, actin cables and "worms" were completely absent from these cells (S29 Movie) (Fig 16C), which argues against the possibility that these linear F-actin structures "on the move" could "prime" Arp2/3, at least when Dip1 is present. To address the hypothetical role of formin when Dip1 is absent, we constructed a *sepA1 dip1Δ* double mutant expressing Lifeact-GFP. Fig 16D shows a temperature-shift experiment with double mutant cells. At 28°C the actin cytoskeleton looked similar to that in single *dip1Δ* cells, although both the SPK and actin cables emanating from it appeared fainter than in the double mutant. When these double mutant cells were shifted to 37°C and incubated at this temperature far beyond the time required to disassemble the SPK, the distribution of actin resembles that observed in the *sepA1* mutant under similar conditions (Fig 16D), *i.e.* actin patches distributed across the hyphae and complete absence of cables or "worms". Kymographs established that these patches "cycled" normally, indicating that they not represent the remnants carried over from the permissive incubation period used to pre-culture hyphae (Fig 16E). Lastly, these cells were still viable, as established by the fact that conspicuous formation of linear F-actin structures was reestablished after shifting them down to the permissive temperature. Therefore, our data argue against the possibility that a formin-mediated mechanism could partially compensate the actin patch-seeding deficit resulting from *dip1Δ*.

## Discussion

The process of endocytosis is intimately associated with the mode of life of filamentous fungi, which form long tubular cells, denoted hyphae. Hyphal tip growth is crucially dependent on the polarization of cell wall modifying enzymes, which is mediated by exocytosis and endocytic recycling [7,36]. Therefore, hyphal tip cells are arguably among those which are most dependent on endocytosis for viability. Studies in the budding and in fission yeasts have revealed the involvement of a highly conserved pathway in which different protein modules act sequentially to drive invagination of the primary endocytic vesicle against internal turgor [12]. Parallel work with metazoan cells revealed that the core machinery is strongly conserved across the lineage that includes fungi and Metazoa [14]. Remarkably, in spite of the applied interest of filamentous fungi as producers of enzymes and metabolites, or of their clinical interest as devastating pathogens, the process of

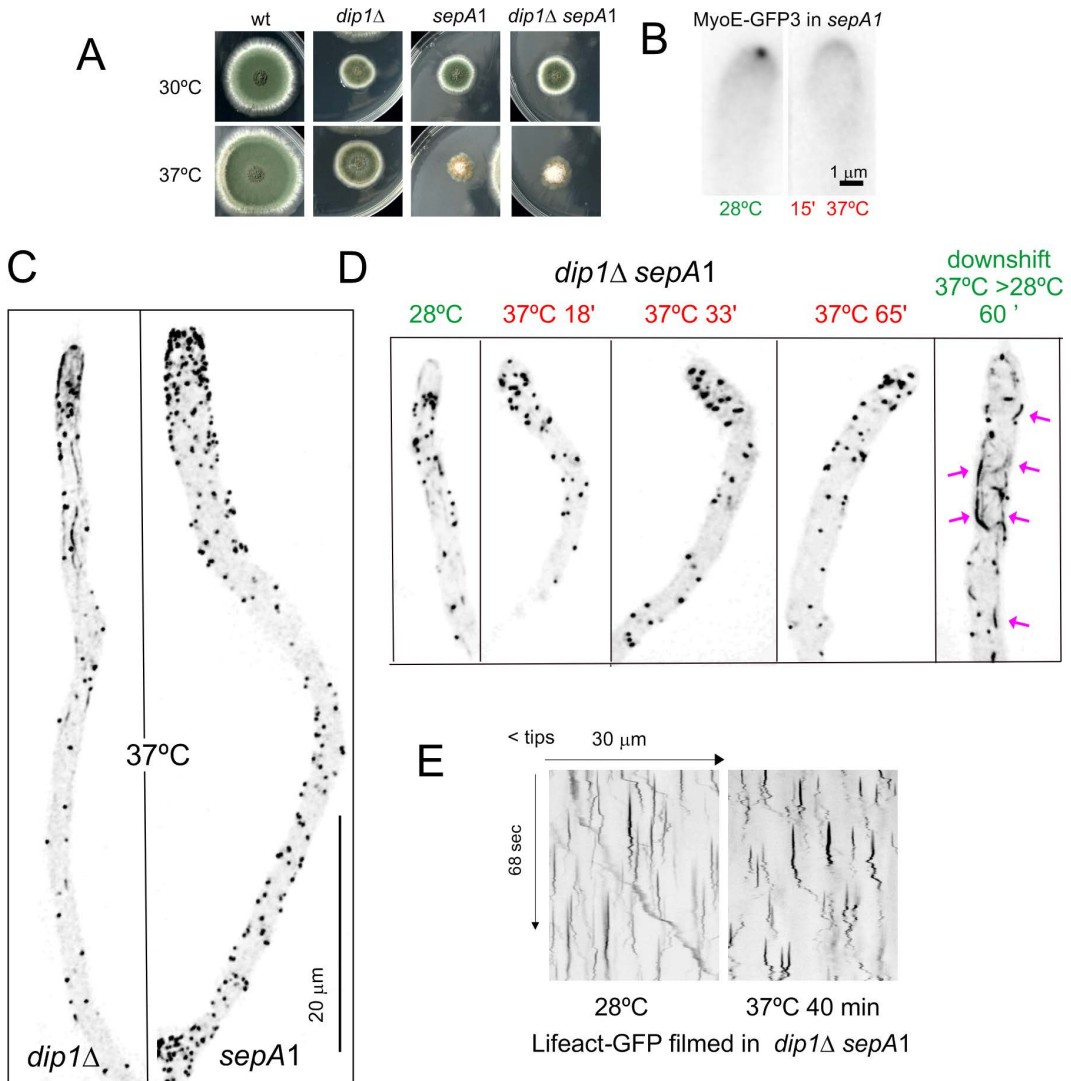

**Fig 16. Actin cables and actin worms are not detected in *sepA1* cells incubated at the restrictive temperature.** (A) *sepA1* and *did1Δ* do not show a synthetic negative growth interaction. (B) *sepA1* delocalizes MyoE from the SPK following a 15 min incubation at 37°C **(C)** In cells incubated at the restrictive temperature, *sepA1* prevents the formation of actin cables or actin worms. Images are MIPs of deconvolved Z-stack stacks. (D) *sepA1 did1Δ* double mutant cells contain actin patches even after one hour of incubation at 37°C. (E) Actin patches in *sepA1 did1Δ* double mutant cells incubated at 37°C cycle normally.

endocytosis requires a better understanding even in model organisms such as *N. crassa* and *A. nidulans*. One of the aims of this study is to fill this gap, approaching endocytosis as one of the major physiological functions of the actin cytoskeleton. A second study on the role of F-actin in exocytosis will be reported elsewhere.

We used three different parameters to determine the extent to which a genetic intervention impairs endocytosis. One was the time of residence of F-actin in endocytic patches. The second was the colony phenotype because, due to the role of endocytosis in hyphal growth, the greater the impairment of endocytosis is, the more debilitating that the causative mutation becomes for colony growth. The third was the efficiency of polarization of the chitin synthase ChsB, a model cargo which is a substrate of the endocytic recycling pathway. The time of residence of F-actin in patches was determined

with a time resolution of 0.2 seconds using Lifeact, chromobody and tractin, tagged with FPs. This parameter was highly reproducible in the wild-type, independently of the probe and FP used, with an average value of 13–14 seconds at 25–28°C. Of note, endogenously FP-tagged proteins belonging to the actin module showed consistent lifetimes (of 13.2, 13.1, 13.9 and 12.9 sec for AbpA[Abp1], FimA, ArpC1 and actin, respectively) buttressing this conclusion. In contrast, each and every mutation that had a negative impact on colony growth extended the time of residence of actin in patches in a roughly commensurate manner, with the largest value (34 sec) resulting from the severely debilitating synthetic negative interaction between *capAΔ* and *abpAΔ*, predictably reflecting a marked deficit in barbed end capping activity. *srv2Δ, dip1Δ, vrp1Δ,* and *capAΔ*, which resulted in less severe colony growth defects, showed F-actin resident times approximately doubling the wild type. *fimAΔ, arpC1Δ, slaBΔ* and *cof1Δ* resulted in lethality. Spores carrying a lethal mutation can be rescued in heterokaryosis and allowed to germinate until they reach a terminal phenotype. In all these three cases, mutant strains were able to establish polarity but arrested growth shortly after that, with a swollen tip phenotype that resembles the already described *slaBΔ* phenotype [6], indisputably establishing that endocytosis is required to maintain polarity during hyphal tip growth. The lethal phenotype of *fimAΔ* deserves separate consideration, because it served to establish that the previously described *fimA::Tn341* mutation [4] is not a complete-loss-of-function allele, which was instrumental to reveal that when fimbrin is compromised, F-actin patches do not transit from the non-motile to the motile phase, a phenotype that this mutation shares with *vrp1Δ*. Vrp1 has been reported to regulate NPF factors, and we show here that it arrives to patches at the beginning of the F-actin cycle.

An important observation was that actin patches frequently overlap with actin cables, an observation previously reported with *N. crassa* and *Trichoderma atroviride* [22,79]. In many examples time-lapse series strongly suggest that patches develop in close association with cortical cables (Figs 4C and S9E and S7 Movie). However, in numerous examples actin patches appeared to give rise to short linear actin structures (Fig 4D and S20 Movie) that often grew to form longer cables (Fig 5 and S12 Movie). These longer cables were eventually fragmented, in all likelihood by F-actin severing proteins, and the resulting actin "chunks" undergo movement, generally in basipetal direction, with a characteristic "crawling" that led us to denote these structures as "actin worms" (Fig 4E and S9–S12 Movies), which became abundant in the *srv2Δ* mutant (see S21 Movie). Our experiments strongly suggest that these actin "worms" are different from exocytic actin cables. Exocytic cables originate at the SPK, are decorated with tropomyosin and appear to be disassembled as they grow away from the tip. In contrast, "actin worms" apparently emerge from actin patches, do not contain tropomyosin, and are able to move for considerable distances, generally away from the tip, before disappearing. These "worms" have the potential to redistribute G-actin across the whole length of the hyphal tip cell.

The close association of patches and cortical cables would be consistent, *a priori*, with the latter serving as seeds for the Arp2/3 complex-mediated burst of actin polymerization, as suggested [78]. This possibility led us to consider the origin of the seed filaments which are required by Arp2/3. *A. nidulans* has a single formin, SepA. Prompted by the observation that inactivation of this protein does not appear to have any consequence in the biogenesis of actin patches or in the duration of the burst of F-actin actin in these structures, making its direct participation in this processes very unlikely, we explored the physiological role of the WISH/DIP/SPIN90 family protein Dip1. Dip1 binds and activates Arp2/3 without any pre-existing seed, giving rise to linear filaments [15–17,75,76]. Ablation of *A. nidulans* Dip1 resulted in major changes in the organization of the actin cytoskeleton. Firstly, the endocytic collar was disorganized, but actin patches were still present. Secondly, these patches were formed at a considerable distance from the apical dome. Thirdly, the SPK was not disorganized and time-lapse sequences revealed that it was the origin of waves of actin cables that under these mutant conditions became, conspicuously visible. Notably, the fact that *dip1Δ* cells are able to generate actin patches establishes that there must be (a) mechanism(s) alternative to Dip1 that mediate the biogenesis of endocytic actin structures when this protein is absent.

We addressed the possibility that such a mechanism is mediated by a formin function that would be masked when Dip1 is present and revealed only in the absence of the latter. Our analysis of *dip1Δ sepA1* double mutant cells argues against

this possibility. These cells are still able to generate actin patches after having been shifted to restrictive conditions. This observation and the fact that cells in which the formin has been inactivated are devoid of actin cables and actin "worms" argues against the possibility that these large linear actin structures provide the mechanism alternative to Dip1. However, we note that remnants of these structures, resulting, for example, from the action of cofilin, could provide microfilaments that could escape detection and that would serve as seeds for Arp2/3 by a sort of "sever, diffuse and trigger" mechanism similar to the one proposed in *S. pombe* [78]. Whatever the nature of this alternative mechanism(s), it must account for the observation that in the absence of Dip1 the endocytic collar is completely disorganized and the apical dome and the subtending region are free of actin patches, suggesting that this alternative mechanism(s) cannot work efficiently near the tip.

The fact that actin cables emanating from the SPK are decorated with tropomyosin might have important implications with regard to the organization of the actin collar, as it is likely that exocytic and endocytic F-actin localizing to the tip and environs compete for the pool of G-actin. It is important to note that tropomyosin outcompetes fimbrin from formin-derived actin cables whereas fimbrin outcompetes tropomyosin from Arp2/3-derived actin networks [80,81]. The facts that tropomyosin is strongly concentrated in the SPK (Fig 1C), that the SPK F-actin network is dependent on the activity of the SepA formin (Fig 14B), and that it does not contain Arp2/3-branched actin (Fig 11) might suggest that Arp2/3 is outcompeted by tropomyosin also in this locale.

## Materials and methods

### *Aspergillus* techniques

*Aspergillus* strains were grown on standard culture media [82]. Cassettes to generate fluorescent proteins or null mutants were introduced into the appropriate strains by transformation and homologous recombination [83] using *nku*Δ recipient strains [84]. All strains were tested by PCR using oligonucleotides flanking the cassettes to ensure correct cassette integration. The full list of strains with their genotypes is listed in S1 Table.

### Null mutant strains and fluorescent protein tagging

All deletion mutations used in this work, namely *arpC1*Δ, *vrp1*Δ, *fimA*Δ, *srv2*Δ, *dip1*Δ, *cofA*Δ, *capA*Δ [73] and *abpA*Δ [6], were constructed by transformation and gene replacement using cassettes generated by fusion PCR with appropriate selection markers. *niiAp::SlaB*<sup>Sla2</sup> is a conditional expression allele, in which SlaB<sup>Sla2</sup> (also denoted SlaB) is expressed under the control of the nitrate-inducible, ammonium-repressible nitrite reductase promoter [24]. *fimA::Tn341* is a transposon insertion 341 pb downstream from the start codon of *fimA* [4]. *sepA1* is a termosensitive allele encoding a mutant formin carrying a Leu1639Ser substitution. This residue lies in the conserved FH2 domain [68].TpmA-tdT, Lifeact-GFP/tdT, actin chromobody-mChFP/TAgGFP, Tractin-GFP and *Aspergillus* actin-GFP were expressed under the control of *inuA* promoter. These cassettes were integrated into the *inuA* locus [47]. ArpC1-GFP, SepA-3xGFP, Vrp1-GFP, Srv2-GFP, Dip1-GFP, CapA-GFP, Cof1-GFP, FimA-mChFP, FimA-GFP, AbpA<sup>Abp1</sup>-GFP/mRFP, SlaB<sup>Sla2</sup>-GFP, MyoE-3xGFP, MyoE-mChFP, SagA<sup>End3</sup> were C-terminally tagged endogenously. ChsB was N-terminally tagged endogenously [7]. All cassettes were generated by fusion PCR using the primers listed in S2 Table.

### Details of F-actin reporters driven by the *inuA* promoter

All reporter proteins/peptides were separated from the fluorescent protein moieties by a (Gly-Ala)5 linker. Lifeact transgenes contain the 17 N-terminal amino acids of Abp140 from *S. cerevisiae* (MGVADLIKKKFESISKEE) fused to GFP or tdT. Actin chromobody transgenes were constructed using the commercial plasmid Actin-Chromobody TagGFP2 (ChromoTek #ACG) as template for PCR amplification of the coding region of α-actin nanobody. This fragment was used to generate by fusion PCR the TagGFP or mChFP cassettes with appropriate selection markers. The tractin reporter

is composed of residues 10–52 of the rat inositol 1,4,5-triphosphate 3-kinase A (ITPKA) protein [31]. The DNA fragment coding for this reporter was synthesized by GenScrip (https://www.genscript.com/) and translationally fused by its C-terminus to the GFP coding region. The GFP-actin transgene expressed under the control of the *inuA* promoter was integrated by gene replacement at the *inuA* locus (inulinase is required only if *Aspergillus* is cultured on inulin as the only carbon source). The strain generated has a backup untagged copy of the actin gene. The TpmA-tdTomato cassette was integrated in the *wA* locus and expressed under its own promoter. The strain generated has a backup untagged copy of TpmA.

## Western blots

For quantitation of Lifeact expression levels in cells cultured on different glucose and sucrose concentrations, the Lifeact-GFP strain was grown overnight at 30°C in minimal medium (MMA) containing the different carbon sources indicated on Fig 1B. Total protein extraction has been described [85]. The following antibodies were used: Lifeact-GFP; Merck'#11814460001 monoclonal anti-GFP as primary antibody (1/5000) and HRP-conjugated AffiniPure goat anti-mouse IgG (H+L) secondary antibodies (Jackson Immunoresearch #115-035-003, 1/5000). For tubulin, anti α-tubulin (Merck # T9026) monoclonal as primary antibody and HRP-conjugated AffiniPure goat anti-mouse IgG (H+L) as secondary antibodies (Jackson Immunoresearch #115-035-003, 1/5000). Reacting bands were detected with Clarity western ECL substrate (Biorad Laboratories #1705061).

## Fluorescence microscopy

*Aspergillus* strains were cultured "on stage" in Ibidi 8-well μ-slide uncoated chambers (Ibidi Cat#80821), using "watch minimal medium" (WMN) [23] with 0.1% glucose as carbon source when expression was driven by the endogenous promoter, or with a mixture of 0.05% glucose and 0.025% sucrose (w/v in all cases) when expression of the reporter genes was driven by the *inuA* promoter. All wide-field imaging was carried out in Leica DMi6 or DMi8 inverted microscopes. The details of the software, hardware, image acquisition, analysis and presentation have been thoroughly described [67]. Deconvolution was carried out with Huygens software (Scientific Volume Imaging B.V.). Benomyl was used at 5 μg/ml, and latrunculin B at 50-100 μM.

## Quantitation of the half-life of actin patches

For Movies acquired at 5 fps, individual frames were acquired with Hamamatsu ORCA-ER CCD or Hamamatsu Flash CMOS cameras equipped, for dual color imaging, with DV2 (Mag Biosystems) and Gemini (Hamamatsu) beam splitters. Time series were usually acquired at 5 fps and streamed to the memory of the computer using Metamorph software. Chimeric reporters usually allowed the acquisition of 600–900 frames (i.e.,. 2-2.5 min of coverage) before hyphae were photo-damaged. When needed, exposure times were increased and the total number of frames modified accordingly to cover equivalent overall times. For example, for proteins residing in patches for long times, as is the case of SlaB[Sla2], or for certain mutants resulting in long lifetimes of Lifeact-GFP in patches, time series were acquired at 1 fps. Time-lapse series were analyzed with kymographs, using as ROIs lines traced across the longitudinal axis of the hyphae. These ROIs were usually wide enough to cover between half and the whole width of the hyphae. In these kymographs, actin patches gave rise to approximately vertical traces consisting of a static phase which was followed by a wiggly phase in which patches underwent movement along the longitudinal axis before disappearing. The actual length of a patch lifetime was determined by estimating the length of the vertical line in pixels, using the region measurement tools of Metamorph software, and converting this length into time units (one pixel usually equivalent to 0.2 sec). In order to avoid errors, we considered only well-separated patches and, when overlapping patches required to be resolved, we converted x,y,t stacks into x,y,z stacks that were visualized in 3D with Imaris Viewer 3.5.1. Rotation of these 3D views usually facilitated resolution of

overlapping patches. Statistical analysis and graphical representation of data were carried out with GraphPad Prism software (v. 7.03).

### Endocytic recycling assays

Endocytic recycling polarizes cargoes such as ChsB. Therefore, we took the distance to which the transmembrane protein ChsB diffuses away from the apex as an indirect measurement of the efficiency of endocytosis. This distance was calculated with a line drawn with the "regions measurement" menu of Metamorph from the apex to the most basal point to which GFP-ChsB reached.

### Confocal imaging

We used a Leica TCS SP8 scanning microscope equipped with a "white light" laser enabling precise excitation wavelength selection. In typical z-stack acquisitions such as that shown on Fig 11C, we used a 63x immersion oil objective, with x,y,z dimensions of 50, 50, 200 nm at 1 AU, laser settings at 483 nm for the GFP channel and 585 nm for the mChFP channel, and standard PMT detection. Raw images were digitalized at 16 bits and manipulated using LAS X software (Leica Microsystems GmbH).

### Supporting information

**S1 Fig. Tropomyosin is essential for viability and TpmA-GFP is not functional.** (A) Heterokaryon rescue of *tpmAΔ*. Heterokaryotic strains carrying *pyrG89 tpmA+* and *pyrG89 tpmAΔ::pyrG^{Af}* nuclei were generated by transformation. Uninucleate, haploid spores derived from these strains were able to grow when supplemented with pyrimidines, but not in their absence, as the ability to grow without them is linked to a lethal *tpmAΔ::pyrG^{Af}* allele (*pyrG^{Af}*. Is the *A. fumigatus pyrG* gene complementing *pyrG89*). Left, replicas of several of these strains in media supplemented or not with pyrimidines. Right, PCR genotyping of three such strains using *tpmA* flanking primers demonstrating the presence of both *tpmA+* and *tpmAΔ::pyrG^{Af}* in them. (B) *tpmAΔ::pyrG^{Af}* conidiospores were able to established polarity but arrested growth shortly after, showing morphogenetically defective curly hyphae. (C) Heterokaryon rescue test as above showing that strains carrying *tpmA* endogenously tagged with GFP are unable to grow as homokaryons.
(TIF)

**S2 Fig. Mutual dependence of the actin and microtubule cytoskeletons.** (A) Frames of a time-lapse sequence showing numerous instances in which MTs reach the SPK and its intermediate neighborhood. MTs are labeled with GFP-tagged alpha tubulin whereas actin was visualized with Lifeact-tdT. Images were deconvolved and further contrasted using the unsharp mask filter of Metamorph (B) A hyphal tip cell expressing GFP-tagged alpha-tubulin that had been incubated for 60 minutes with 50 μM latrunculin B. Contacts of the plus-ends of MTs with the cortex, indicated with orange arrows, are distributed all across the apical dome. (C) Recovery of the actin cytoskeleton after having been disorganized by depolymerization of MTs with benomyl (5 μM). The first frame of the movie corresponds to the six minute timepoint after washing out the drug. Top left, MIPs of Z-stacks back acquired at the indicated time points after washing out benomyl. Top, right, kymograph obtained with a longitudinal line ROI covering the whole width of the hypha, with deep blue representing the strongest signal and green the weakest. Changes in the rate of apical extension are noted. Bottom, individual sections of the Z-stacks used for MIPs are shown. These data have been extracted from S3 Movie.
(TIF)

**S3 Fig. Tools to determine actin patch lifetimes with longitudinal kymographs.** (A) Kymographs were usually obtained with linear ROIs traced longitudinally across the hyphae. ROIs were usually 12–25 pixel wide. If, as shown in this example, traces of individual endocytic events excessively overlapped, time stacks were manipulated as Z-stacks using

Imaris software, which facilitated detection of individual traces. Details are also provided under Materials and Methods. (B) Examples of transverse kymographs.
(TIF)

**S4 Fig. Dynamics of the endocytic coat module component SagA^End3.** (A) middle planes of deconvolved Z-stacks showing that SagA^End3 localizes to endocytic patches but not to the SPK. (B) and (C) Kymograph analysis of the time of residence of SagA^End3 in endocytic patches. Error bars represent the means±S.D. (D) Region of a kymograph obtained with a strain coexpressing GFP-tagged SagA^End3 and Lifeact tdT. (E) Example of an actin patch from the above strain.
(TIF)

**S5 Fig. Impaired endocytic recycling of GFP-SynA in *fimA::Tn341* cells.** (A) MIP and corresponding Z-stack of a wild-type hyphal tip cell expressing GFP-SynA. (B) As in (A) for a *fimA::Tn341* cell. (C) Morphological transitions following germination of a *fimA::Tn341* conidiospore. (D) Uniform distribution of GFP-SynA in a markedly abnormal *fimA::Tn341* germling. MIP. (E) Individual sections of the region boxed in (D) showing the absence of any remnant of SPK. (F) Lifeact-GFP distribution in a *fimA::Tn341* cell which is morphogenetically normal compared to the wild type. (G) After a period of normal morphology *fimA::Tn341* hyphae became swollen in the tips and arrested growth. (F) MyoE was completely delocalized to the cytosol in these *fimA::Tn341* hyphae incubated for very long periods.
(TIF)

**S6 Fig. Cofilin localization in a heterozygous diploid.** (A) MIP of a deconvolved Z-stack of Cof1-GFP expressed in a heterozygous diploid with the wild-type allele to maintain viability. (B) Analysis of Cof1-GFP endocytic patches. Error bars represent the mean±S.D.
(TIF)

**S7 Fig. Phenotypic aspects of *srv2*.** (A) Ablation of Srv2 results in disorganization of the actin collar. (B) The SPK is disorganized in *srv2Δ* cells. MyoE- mChFP and Lifeact-GFP distribution. MyoE forms a crescent in which individual dots can be resolved. Images are successive planes of z-stack separated by 0.25 μm in the axial dimension. (C) Localization of endogenously tagged Srv2-GFP. Note that the protein is not fully functional. (D) Lifetime of actin patches of endogenously tagged Srv2-GFP and comparison with Lifeact-GFP patches in the wild-type. (E) "Worms" of F-actin moving towards the base of an *srv2Δ* strain, with a kymograph showing the corresponding speeds.
(TIF)

**S8 Fig. Phenotypic aspects of *capA*.** (A) Growth tests on minimal medium comparing the wild-type with a strain expressing endogenously tagged CapA-GFP and the corresponding *capAΔ* deletion mutant, showing that CapA-GFP is only partially functional. (B) Average lifetime of CapA-GFP in cortical patches. Error bars represent the mean±S.D. (C) Middle planes of hyphal tips with the indicated genotypes showing the localization of the recycling cargo GFP-ChsB. Asterisks indicate the SPK, which is markedly less prominent when CapA was ablated, as indicated with a smaller asterisk. The SPK is not detectable in the *abpAΔ capAΔ* double mutant. (D) Split tip in a *capAΔ* strain expressing Lifeact-GFP. (E) Kymograph of Lifeact-GFP in the *abpAΔ capAΔ* double mutant showing that actin patches are static and do not mature. (F) Multiple polarity axes in an *abpAΔ capAΔ* double mutant cell expressing Lifeact-GFP.
(TIF)

**S9 Fig. Phenotypic aspects of *dip1*.** (A) Growth test showing that Dip1-GFP is functional. (B) The time of residence of GFP-tagged fimbrin in endocytic patches is nearly duplicated in *dip1Δ* cells, closely resembling the behavior of Lifeact-GFP. (C) Comparison of apical extension rates of the wild-type and of the *dip1Δ* mutant. The apical extension rate of *dip1Δ* tips is half of the wild-type. Error bars represent the means±S.D. (D) Deconvolved middle plane of a *dip1Δ* hypha expressing Lifeact-GFP. (E) Example of an actin filament elongating throughout the sequence of 53 seconds (its leading

end is indicated with a red dot). Some actin patches appeared to develop associated with the filament, whereas other appear to be captured by the growing end.
(TIF)

**S10 Fig.  Tropomyosin only labels those actin cables emerging from the SPK.**
(TIF)

**S1 Table.  List of strains used in this work.**
(PDF)

**S2 Table.  List of oligonucleotides used in this work.**
(PDF)

**S1 Movie.  Distribution of actin structures labeled with Lifeact-GFP over a period of one hour.** Time is in min:sec:msec.
(MP4)

**S2 Movie.  Dual channel 4D acquisition (0.5 fpm) of nuclei/chromatin and F-actin/Lifeact.** Time is in hs:min.
(MP4)

**S3 Movie.  4D (x,y,z,t) time series (1 fpm) showing the recovery of the actin collar and hyphal tip growth after ben-omyl treatment and subsequent washout.** Time is in hs:min.
(MP4)

**S4 Movie.  3D (x,y,t) series of GFP-actin acquired at 5 fps.** Time is in sec:msec.
(MP4)

**S5 Movie.  3D (x,y,t) series of Lifeact-GFP acquired at 5 fps.** Time is in sec:msec.
(MP4)

**S6 Movie.  3D view of a time stack of Lifeact-GFP that has been treated as a z-stack to facilitate the visual separa-tion of individual patches that otherwise overlap in 2D.**
(MP4)

**S7 Movie.  3D time series of Lifeact-GFP acquired at 5 fps, focusing on a peripheral actin cable revealing the for-mation of patches associated with it.** Time is in sec:msec.
(MP4)

**S8 Movie.  3D time series (5 fps) of Lifeact-GFP that has been contrasted to reveal the presence of filamentous connections between patches.** Time is in sec:msec.
(MP4)

**S9 Movie.  4D time series of Lifeact-GFP at 0.75 fps depicting "actin worms" of F-actin moving basipetally.** Time is in min:sec.
(MP4)

**S10 Movie.  4D time series of Lifeact-GFP filmed at 37°C and 0.2 fps depicting "actin worms" of F-actin moving basipetally.** Time is in min:sec.
(MOV)

**S11 Movie.  4D time series of Lifeact-GFP filmed at 28°C and 0.2 fps depicting "actin worms" of F-actin moving basipetally and acropetally.** Time is in min:sec.
(MP4)

**S12 Movie. 3D time series of a hyphal tip expressing Lifeact-GFP and photographed at 1 fps.** An actin cable appears to be generated with F-actin originating from the SPK and from the foremost actin patches. The cable subsequently gave rise to worms.
(MP4)

**S13 Movie. Three dimensional reconstitution of a z-stack of AbpA[Abp1]-GFP hyphae.**
(MP4)

**S14 Movie. Abortive germling derived from a *slaB*-deficient conidiospore showing the characteristic Lifeact-GFP actin patches that do not progress over time.** The Movie was acquired at 4 fps.
(MP4)

**S15 Movie. Recovery of a germling carrying an conditionally-expressed *slaB* allele shifted from repressing to inducing conditions (ammonium and nitrate as sole nitrogen sources, respectively).** Frames are maximal intensity projections of z-stacks.
(MOV)

**S16 Movie. Maximal intensity projection and individual optical sections of a deconvolved z-stack of Lifeact-GFP in an abortive *fimA*Δ germling.**
(MP4)

**S17 Movie. Composition of 3D (x, y, t) Movies (middle planes, 5 fps) depicting the behavior of F-actin patches in a *fimA::Tn341* hypha compared to the wild-type.** Time is in sec:msec.
(MP4)

**S18 Movie. Behavior F-actin patches in a *srv2*Δ hyphal tip filmed with Lifeact-GFP at 5 fps.** Time is in sec:msec.
(MP4)

**S19 Movie. Numerous "actin worms" in a *srv2*Δ cell filmed at 1 fps.** Time is in min:sec.
(MP4)

**S20 Movie. Mutant *srv2*Δ tip with cortical patches giving rise to actin "actin worms" that move away and towards the apex (on the right).** 3D time series at 0.5 fps. Time is in min:sec.
(MP4)

**S21 Movie. A 'worm' apparently composed of two parallel cables originates at an actin patch and moves basipetally.** Lifeact-GFP photographed every 200 msec. Time is in sec:msec.
(MP4)

**S22 Movie. ArpC1-GFP filmed at 5 fps.** Time is in sec:msec.
(MP4)

**S23 Movie. ArpC1-mChFP and Vpr1-mChFP co-filmed at 0.2 fps.** Time is in min:sec.
(MP4)

**S24 Movie. Time series of Lifeact-GFP in *capA*Δ *abpA*Δ cells acquired at 10 fps.** Time is in sec:msec.
(MOV)

**S25 Movie. Time series (1 fps) of a *dip1*Δ cell showing that actin cables are continuously being generated at the SPK.** Time is in sec.
(MP4)

**S26 Movie. Time series (1 fps) showing conspicuous actin cables labeled with Lifeact-GFP in a *dip1*Δ cell.** Time in min:sec.
(MP4)

**S27 Movie. Conspicuous actin cables decorated with tropomyosin-GFP originating from the SPK of a *dip1*Δ cell and filmed at 5 fps.**
(MOV)

**S28 Movie. Tropomyosin-GFP cells filmed at 1 fps, showing that tropomyosin is not present in worms or in actin cables that do not originate at the SPK.**
(MP4)

**S29 Movie. Actin worms and actin cables are completely absent from *sepA1* cells that had been shifted to 37°C.** This time series was acquired at 5 fps.
(MP4)

## Acknowledgments

We thank Brian Shaw, Xin Xiang and Berl Oakley for generously sharing strain and to Sara Abib for skillful technical assistance.

## Author contributions

**Conceptualization:** Mario Pinar, Eduardo A. Espeso, Miguel A. Peñalva.

**Data curation:** Mario Pinar, Paula Polonio, Eduardo A. Espeso.

**Formal analysis:** Mario Pinar, Paula Polonio, Eduardo A. Espeso, Miguel A. Peñalva.

**Funding acquisition:** Eduardo A. Espeso, Miguel A. Peñalva.

**Investigation:** Marisa Delgado, Mario Pinar, Paula Polonio, Sergio Fandiño, Miguel A. Peñalva.

**Methodology:** Marisa Delgado, Mario Pinar, Paula Polonio, Miguel A. Peñalva.

**Resources:** Marisa Delgado.

**Supervision:** Mario Pinar, Eduardo A. Espeso, Miguel A. Peñalva.

**Visualization:** Mario Pinar, Paula Polonio, Miguel A. Peñalva.

**Writing – original draft:** Mario Pinar, Miguel A. Peñalva.

**Writing – review & editing:** Mario Pinar, Miguel A. Peñalva.

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
