## [Decision Letter · Decision Letter 0]

31 Mar 2025

PGENETICS-D-25-00167

The actin module of endocytic internalization in Aspergillus nidulans: a critical role of the WISH/DIP/SPIN90 family protein Dip1

PLOS Genetics

Dear Dr. Peñalva,

Thank you for submitting your manuscript to PLOS Genetics. After careful consideration, we feel that it has merit but does not fully meet PLOS Genetics's publication criteria as it currently stands. Therefore, we invite you to submit a revised version of the manuscript that addresses the points raised during the review process.

***  Miguel, I apologize sincerely for the prolonged review period.  There were matters outside of anyone's control that required accommodation if we were to have well-qualified reviewers.  In any case, all three reviewers were highly supportive, as you will see from their comments below.  One consistent recommendation was to add some explanations and in some cases details.  The goals were to improve understanding by specialists and to make the study more accessible to non-specialists.  These edits will be straightforward.  There were also some recommendations for additional controls or experiments.  Rev 1 presented some really interesting suggestions, and they seem quite do-able to me, such as the tropomyosin labeling and the phenotyping of dip1∆ sepA double mutants.  See what you think; they have the potential to tighten up the story.  Rev 2 suggested a FM4-64 staining experiment that seems straightforward.  Please give it some thought.  Warmest regards, Aaron***

Please submit your revised manuscript within 60 days May 30 2025 11:59PM. If you will need more time than this to complete your revisions, please reply to this message or contact the journal office at plosgenetics@plos.org. Please include the following items when submitting your revised manuscript:

We look forward to receiving your revised manuscript.

Kind regards,

Aaron P. Mitchell, PhD

Academic Editor

PLOS Genetics

Eva Stukenbrock

Section Editor

PLOS Genetics

Aimée Dudley

Editor-in-Chief

PLOS Genetics

Anne Goriely

Editor-in-Chief

PLOS Genetics

**Journal Requirements:**

- ® on page: 23.

**Reviewers' comments:**

Reviewer's Responses to Questions

**Comments to the Authors:**

Reviewer #1: This manuscript from the Penalva lab describes endocytic actin patches in Aspergillus and covers a large ground. The authors can be commended for their extensive work, high-quality imaging, and clear descriptions, which brings high standards to cell biology work with this filamentous organism. By tagging and deleting key proteins involved in actin assembly and regulation, focusing on regulators of actin patches in yeast, they describe similarities and differences in Aspergillus compared to the well-studied yeast cases. An important focus of the work is on identifying the source of the mother filament for Arp2/3 assembly at endocytic sites. They reveal an interesting actin “worm” structure, which often overlaps with actin patches and is suggested to help new nucleation. They also describe the phenotype of dip1∆, which largely, but not completely, abrogates patch formation.

Overall, I find the work very interesting and well supported. I have two main comments – on methodology and on the dip1-independent source of mother filaments – which I feel need to be addressed before publication. These experiments should be doable with the tools that the authors have at their disposal.

**Major points:**

It is not clear to this reviewer exactly how kymographs are constructed, as this is not clearly explained in the methods. For longitudinal kymographs, used in most of the study, what is the width of the line that is captured? If the line is only drawn at the cell cortex, such kymographs cannot capture the inward movement of actin patches as they invaginate and break off from the plasma membrane, as they would simply disappear from the kymograph. However, it seems that the line becomes then wispy and wiggly, suggesting that part of the inward movement is captured. How much of it? It is also not clear whether the transition to invagination can be clearly established with this analysis strategy, in contrast to the transversal approach used in most of the yeast work. In the examples shown in Fig 3C, taking the transition from a static to a wiggly line as start of invagination, it seems like the residence time at the membrane is much shorter in the transversal than longitudinal examples. These points should be clarified to better establish the analysis method. More generally, there needs to be more precise description of image quantification methods. Currently, all is stated is that "the length of the traces was estimated visually and converted into timescales". This does not suffice for anyone to reproduce the work.

The origin of the actin “worms” is mysterious. The text variously hypothesizes that they originate from the SPK, implying a formin-dependent assembly, or that they initiate at a patch, suggesting elongation of Arp2/3-nucleated filaments. With the tools and mutants at hand, this question should be quite easily addressed: Are the actin "worms" labeled by tropomyosin, as would be likely expected if they are composed of linear actin filaments? Do they form in the sepA mutant?

Linked to the previous issue, the manuscript makes a strong statement that formin is not involved in generating the mother filament for Arp2/3 nucleation. However, the authors show that, although Dip1 plays an important role, there is also a Dip1-independent mechanism. It seems entirely possible to me that this Dip1-independent mechanism relies on formin activity, whose role may be masked in presence of Dip1. To test this, it would be important to examine the phenotype of dip1∆ sepA double mutants. Alternatively, the mother filament may come from another actin patch. This can also be tested by performing transient depolymerization assays with CK-666 and probing the ability of actin patches to regenerate after washout (in dip1∆, or in dip1∆ sepA∆ if the latter still shows actin patches).

The text makes the statement that fimA loss of function disturbs actin at the SPK even though fimbrin does not localize there. Does the SPK still exist in fimA::Tn341? My understanding is that the SPK is a structure that forms for fast hyphal growth. The slow growth of the fimA disruptant may indirectly perturb the formation of the SPK.

**Minor comments:**

Are C-terminally tagged tropomyosins functional, or do they require co-expression of untagged tropomyosin. It would be helpful to make this clearer.

It is not very clear what the semi-structural illustration in Fig 1A shows. It looks like it has a white box covering part of the tropomyosin structure.

Figure 2E is not very clear. If this is to illustrate that the endocytic collar stays at constant length from the apex over time, it would be helpful to use a rainbow color scheme to label time frames instead of black-and-white projection.

It is not clear to this reviewer how the LatB treatment affects microtubules. In Fig 1G, there is one MT ending at the SPK, but other MT seem to end elsewhere. What is the ground to state that MTs ends are disorganized upon actin depolymerization? If the authors want to make this statement, it should be backed up by some quantification.

In slaB conditional depletion, it would be more informative to show a kymograph of actin than a static image (5F), as this just looks like actin patches but does not show what the text states (clumps hanging from the cortex representing unproductive events...).

Line 389: “to” should read “two”

Line 460: The hypothetical statement that hyphal tip growth might resemble protrusion of filopodia is slightly odd. Tip growth is very well understood in fungi, and, besides involvement of formins and barbed end orientation towards the plasma membrane, does not resemble filopodial growth. I would encourage authors to remove this sentence.

Is the naming of verprolin as vpr1 (rather than vrp1 in yeast) a typo? This is throughout in the text, but variable (vpr1 or vrp1) in figures.

Figure S2C is not described in the legends.

Reviewer #2: The actin module of endocytic internalization in Aspergillus nidulans: a critical role of WISH/DIP/SPIN90 family protein Dip1

The paper is a thorough and comprehensive fungal cell biology study that is interesting and important. The conclusions are predominantly well supported by experimental evidence but there are some areas which could be improved. These are summarised below.

• The authors refer to some of their findings as novel when in some instances they have been long known and, indeed, already published.

• Some figures need attention, as listed below.

• Most of the text is written with US English, but there are some words spelled with British English. Please correct for internal consistency.

• If the Chromobody gives “more accurate” numbers for actin patch lifetime, why did the authors choose to use Lifeact-FP instead in so many experiments? What are the advantages/disadvantages of the 5 actin reporters used?

• The authors make claims about the SPK but do not show it. I suggest carrying out FM4-64 staining in the strains labelled with Lifeact-FP in the mutant backgrounds. Or utilise a SPK reporter such as MyoA or SepA.

• The WT vs dip1� model they propose at the end of the paper can be improved. The cartoon is very simplistic, poorly made with spatial/proportion and important inaccuracies. This should be improved.

• Figures’ legends are a little inadequate in my view and could be improved for detail and accuracy

Line 135- The should add the [22] reference for N. crassa.

Line 136- why is it problematic to observe the “exocytic” cables with Lifeact in A. nidulans in contrast to M. oryzae (Li et al., 2020, Plos Pathogens)?

Line 179- For clarity, specify what combination of sucrose+glucose is used in Figure 1C, either on the figure itself or the figure’s legend.

Line 189- It is worth mentioning that 1.3 �m is the distance of the collar from the apex in A. nidulans, but that it is different in other fungi (e.g. N. crassa).

Line 201- More evidence is needed. See below on Figure 2.

Line 202- How long did the hyphae take to recover after the drug wash out in each case?

Line 229- For consistency, I suggest sticking either to collar or ring. Personally, I think collar is widely used in the past, and that way confusion is avoid (CARs).

Line 236- The presence of F-actin in the core of the SPK has been known since many years, what is the novelty? In the text is understood as if it was sort of a new finding.

Line 291- It seems the authors use the words ‘filament’, ‘cable’, ‘mini filament’, and ‘micro filament’, interchangeably, which is incorrect. Cables, like the other actin higher order structures (patches, SATs, CARs) are composed of F-actin. Therefore, the “worms” are not chunks of filaments, but rather chunks of cables. Please correct throughout the text the use of these words.

Line 348- Endocytosis in the subapical collar is unlikely to be clathrin-mediated (Schultzhaus et al., 2017, Mol. Microbiol.; Martzoukou et al., 2017, eLife). Do the authors have any explanation to this? Especially, since they mention that the uncoupling of the clathrin module and the F-actin module is the reason why the mutation of slab is lethal. This should be discussed.

Line 359- The absence of fimbrin in the SPK has been known for a very long time and yet istreated as a notable “novelty” in the text when it is already well established.

Line 370- Is there still a SPK? Please add an image of the mutant with FM4-64 (or labelled with a SPK reporter) to show if this is an event of loss of polarity, what happens to the SPK? Maybe the Lifeact is still associated with the SPK, and this just has a different morphology in the mutant.

Line 389- Typo “to” or make sentence clear.

Line 396- “Cofilin is essential” for? Section title could be more descriptive.

Line 588- Correct to mChFP, instead of mCh. Also correct throughout the text.

Line 600- Do you mean pathways or proteins?

Line 638- Exocytosis should also be mentioned, as it is essential for hyphal tip growth.

Line 646- While endocytosis studies are still scarce in filamentous fungi, I think insufficiently is not the correct word. There have been a number of good efforts to understand this process in the last decade and a half.

Line 672- Are they really yeast-like? It’s just a sick abnormal germling and doesn’t look like yeast to me.

Line 685- The association of patches and cables has been already observed before in N. crassa (Delgado-Alvarez et al., 2010, FGB) and T. atroviride (Garduno-Rosales et al., 2022, FGB), which should be added in the discussion.

Line 686- In the text, authors only called them “worms”, for consistency modify to “actin worms” in all the text.

Line 700- Is it really striking? Why? Please justify.

Line 708- This needs further evidence to be stated that way. Justify.

Line 722- “Filaments/cables” incorrect.

Line 770- tdTomato.

Line 784- Add space between the dot (.) and Reacting.

Line 795- Correct micrograms and micromolar to their short version.

Line 824- Add the camera that was used to capture the images.

Line 902- Please correct the space between Lifeact and the hyphen.

Line 920- More explanation is needed; the kymograph is not very clear for readers here.

Line 936- Doesn’t look like yeast.

Line 943- For some colourblind individuals is actually very hard to distinguish between and cyan. Choose a colour combination that avoids confusion.

Line 975- Consistency “actin collar”.

Line 984- Show FM4-64 image.

Line 1066- Correct “acting”.

Line 1072- Correct capital letter on “Endocytic”.

Line 1073- Correcy capital letter on “In”.

Line 1377- Where is (C)?

Figure 1- Using green/red palette should be avoid to support colourblind individuals. Please correct in all the pertinent figures. Does Lifect not label septation events in A. nidulans? Same for the other fluorophores. Why only showing it with Chromobody?

Figure 2- Please state what MIPs are for the readers as it is the first time using that abbreviation in the paper.

For (F) would it be possible to colour differently the SPK from the patches to distinguish each population on the front image?

For (G) the figure itself is written as y,z, but in the figure legend is written as z,y. Please arrange for consistency. Also, there is no measurement of co-localisation. Calculate Pearson’s or Manders’ coefficient to quantify real co-localisation.

In (I) is the collar really not present? Or is it just “moved to the tip”? A z-stack would probably give better evidence.

In (H-I) what concentrations of the drugs were used? Indicate in figure and/or legend.

In (J) the numbers (time stamps) are in reverse, correct.

Figure 4- In panel (E) where is the tip? Indicate.

(G) is not self-explanatory.

Figure 5- In (A) what is the point of showing the two FP? Pattern is the same.

Figure 6- If the dotted line is accurate and it indicated the hyphal profile, then the hypha could be stressed (swollen tip), showing an endocytic collar further away from the tip than usual. Change image.

On (E) indicate the mutant background in the image.

Figure 7- This should be a supplemental figure.

Figure 8- On (B) the blue letters get lost with the background, please use a more contrasting colour.

On (D) What happens to the SPK? Using FM4-64 would help to resolve this.

Figure 10- In (E) the right panel looks as if it was an offset between the two proteins. Might be better or separate the images and then show the merge.

Figure 13- Images in panel (C) should be stained with FM4-64 to show the SPK. Please correct.

Figure 14- In (C) please correct for consistency to mChFP.

Figure 15- In (J) please indicate where the tip is.

Figure 16- The endocytic patches look almost outside the hyphae. Please improve this figure.

Supplemental figure 3- In (A) why is the WT 37o so irregular in comparison to the one in supplemental figure 4?

Supplemental figure 5- Are those fungi grown at different temperatures? Indicate.

Reviewer #3: The manuscript entitled 'The actin module of endocytic internalization in Aspergillus nidulans: a critical role of the WISH/DIP/SPIN90 family protein Dip1' by Delgado & Pinar and collaborators is an astonishing piece of work in which a large number of proteins involved in the endocytosis of the model filamentous fungus Aspergillus nidulans are characterized. The authors use three criteria: 1) residence of F-actin in endocytic patches, 2) growth, and 3) endocytic recycling efficiency; to increase knowledge about the endocytic process in fungi showing that endocytosis is essential and providing relevant information on how F-actin drives the internalization of endocytic vesicles. Importantly, the work demonstrates a key role of Dip1 in supplying seed filaments for the formation of F-actin branching, although it makes it clear that is not the only one. The experiments are well designed and superbly executed and interpreted. The conclusions are clear and solidly supported. Among all the results, the in vivo fluorescence microscopy studies are simply incredible. For all these reasons I strongly recommend its publication.

I include a series of non-essential suggestions that I believe would contribute to improving the final version of the manuscript:

1) Summary: In my opinion it is too specialized. I think it would be better to format it to make it easier for the general public to understand. Furthermore, it contains abbreviations such as FPs that are not indicated.

2) Colocalization studies in kymographs: The authors perform different colocalization studies using Lifeact (SlaB, ArpC1) or ArpC1 (Vrp1) but others are missing. I believe it would be very interesting to see when the explosion of F-actin occurs in the cycle of other proteins, particularly those that have a very different period.

3) SlaB under the control of nitrate-inducible, ammonium-repressible promoter: I think it would be fantastic and informative to film, if possible, how growth and coupling of the clathrin and F-actin polymerization modules are restored by exchanging ammonium for nitrate…,

Minor points:

1) Line 85: change ‘. in’ to ‘. In’.

2) Line 87: change ‘[4-6] Endocytic’ to ‘[4-6]. Endocytic’.

3) Line 172: change ‘[47] [48]’ to ‘[47, 48].

4) Line 303: change ‘minfilaments’ to ‘minifilaments’.

5) Line 312: change ‘that If these’ to ‘that if these’.

6) Line 388: change ‘to’ to ‘two’.

7) Line 479: change ‘in a narrow a spot’ to ‘in a narrow spot’.

8) Line 486: change ‘(Figure 11B).’ to ‘(Figure 11B),’.

9) Line 920: change ‘supplemental movie 11’ to ‘supplemental movie 9’.

10) Fig 4G: Point out the microfilament moving towards the tip for clarity.

11) Fig 7: I would move this figure to supplementary. I find it odd to see data for mutants that haven't yet been discussed in the text. Another alternative (maybe better) would be to split it up to include the data for each mutant at the point where it's discussed (always including the wt for clarity).

12) Fig 9E: Indicate what the red arrow means.

13) Supp Fig 2: Revise the legend, (C) is missing.

14) Supp Fig 5: Letters are missing in the figure.

15) Results section ‘Actin polymerization occurs on SlaBSla2 patches’: Indicate that AbpA and SlaB are tagged endogenously.

**Have all data underlying the figures and results presented in the manuscript been provided?**

Reviewer #1: Yes

Reviewer #2: Yes

Reviewer #3: Yes

PLOS authors have the option to publish the peer review history of their article (what does this mean? ). If published, this will include your full peer review and any attached files.

**Do you want your identity to be public for this peer review?** For information about this choice, including consent withdrawal, please see our Privacy Policy .

Reviewer #1: No

Reviewer #2: No

Reviewer #3: No

**Figure resubmission:**
---

## [Decision Letter · Decision Letter 1]

14 Jul 2025

Dear Dr Peñalva,

We are pleased to inform you that your manuscript entitled "The actin module of endocytic internalization in Aspergillus nidulans: a critical role of the WISH/DIP/SPIN90 family protein Dip1" has been editorially accepted for publication in PLOS Genetics. Congratulations!

Yours sincerely,

Aaron P. Mitchell, PhD

Academic Editor

PLOS Genetics

Eva Stukenbrock

Section Editor

PLOS Genetics

Aimée Dudley

Editor-in-Chief

PLOS Genetics

Anne Goriely

Editor-in-Chief

PLOS Genetics

Comments from the reviewers (if applicable):

Reviewer's Responses to Questions

**Comments to the Authors:**

Reviewer #2: I am grateful to the authors for their thoughtful responses to my comments and those of the other reviewers. This is a valuable contribution and I am happy to recommend it is accepted in its current form.

**Have all data underlying the figures and results presented in the manuscript been provided?**

Reviewer #2: Yes

PLOS authors have the option to publish the peer review history of their article (what does this mean? ). If published, this will include your full peer review and any attached files.

**Do you want your identity to be public for this peer review?** For information about this choice, including consent withdrawal, please see our Privacy Policy .

Reviewer #2: No

**Data Deposition**

http://datadryad.org/submit?journalID=pgenetics&manu=PGENETICS-D-25-00167R1

**Press Queries**

---

## [Editor Report · Acceptance letter]

PGENETICS-D-25-00167R1

The actin module of endocytic internalization in Aspergillus nidulans: a critical role of the WISH/DIP/SPIN90 family protein Dip1

Dear Dr Peñalva,

We are pleased to inform you that your manuscript entitled "The actin module of endocytic internalization in Aspergillus nidulans: a critical role of the WISH/DIP/SPIN90 family protein Dip1" has been formally accepted for publication in PLOS Genetics! Your manuscript is now with our production department and you will be notified of the publication date in due course.

With kind regards,

Anita Estes

PLOS Genetics

On behalf of:
